# Do As I Can, Not As I Say:
# Grounding Language in Robotic Affordances

[1] **Michael Ahn**\*, **Anthony Brohan**\*, **Noah Brown**\*, **Yevgen Chebotar**\*, **Omar Cortes**\*, **Byron David**\*,
**Chelsea Finn**\*, **Chuyuan Fu**†, **Keerthana Gopalakrishnan**\*, **Karol Hausman**\*, **Alex Herzog**†,
**Daniel Ho**†, **Jasmine Hsu**\*, **Julian Ibarz**\*, **Brian Ichter**\*, **Alex Irpan**\*, **Eric Jang**\*,
**Rosario Jauregui Ruano**\*, **Kyle Jeffrey**\*, **Sally Jesmonth**\*, **Nikhil J Joshi**\*, **Ryan Julian**\*,
**Dmitry Kalashnikov**\*, **Yuheng Kuang**\*, **Kuang-Huei Lee**\*, **Sergey Levine**\*, **Yao Lu**\*, **Linda Luu**\*,
**Carolina Parada**\*, **Peter Pastor**†, **Jornell Quiambao**\*, **Kanishka Rao**\*, **Jarek Rettinghouse**\*,
**Diego Reyes**\*, **Pierre Sermanet**\*, **Nicolas Sievers**\*, **Clayton Tan**\*, **Alexander Toshev**\*,
**Vincent Vanhoucke**\*, **Fei Xia**\*, **Ted Xiao**\*, **Peng Xu**\*, **Sichun Xu**\*, **Mengyuan Yan**†, **Andy Zeng**\*

\*Robotics at Google, †Everyday Robots

**Abstract:** Large language models can encode a wealth of semantic knowledge about the world. Such knowledge could be extremely useful to robots aiming to act upon high-level, temporally extended instructions expressed in natural language. However, a significant weakness of language models is that they lack real-world experience, which makes it difficult to leverage them for decision making within a given embodiment. For example, asking a language model to describe how to clean a spill might result in a reasonable narrative, but it may not be applicable to a particular agent, such as a robot, that needs to perform this task in a particular environment. We propose to provide real-world grounding by means of pretrained skills, which are used to constrain the model to propose natural language actions that are both feasible and contextually appropriate. The robot can act as the language model's "hands and eyes," while the language model supplies high-level semantic knowledge about the task. We show how low-level skills can be combined with large language models so that the language model provides high-level knowledge about the procedures for performing complex and temporally extended instructions, while value functions associated with these skills provide the grounding necessary to connect this knowledge to a particular physical environment. We evaluate our method on a number of real-world robotic tasks, where we show the need for real-world grounding and that this approach is capable of completing long-horizon, abstract, natural language instructions on a mobile manipulator. The project's website, video, and open source can be found at `say-can.github.io`.

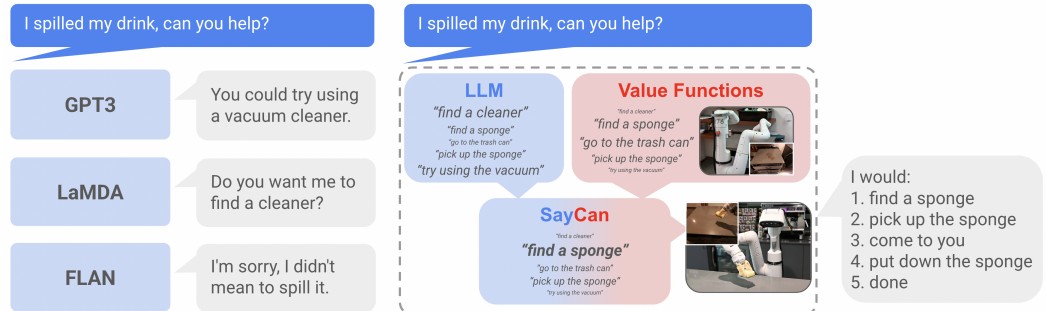

Figure 1: LLMs have not interacted with their environment and observed the outcome of their responses, and thus are not grounded in the world. SayCan grounds LLMs via value functions of pretrained skills, allowing them to execute real-world, abstract, long-horizon commands on robots.

## 1  Introduction

Recent progress in training large language models (LLMs) has led to systems that can generate complex text based on prompts, answer questions, or even engage in dialogue on a wide range of topics. These models absorb vast quantities of knowledge from text corpora mined from the web, and we might wonder whether knowledge of everyday tasks that is encoded in such models can be

---

[1]Authors listed in alphabetical order. Contributions in Appendix A.
Corresponding emails: {`ichter,xiafei,karolhausman`}@google.com.
6th Conference on Robot Learning (CoRL 2022), Auckland, New Zealand.

used by robots to perform complex tasks in the real world. But how can embodied agents extract and harness the knowledge of LLMs for physically grounded tasks?

This question poses a major challenge. LLMs are not grounded in the physical world and they do not observe the consequences of their generations on any physical process [1]. This can lead LLMs to not only make mistakes that seem unreasonable or humorous to people, but also to interpret instructions in ways that are nonsensical or unsafe for a particular physical situation. Figure 1 shows an example – a kitchen robot capable of executing skills such as "pick up the sponge" or "go to the table" may be asked for help cleaning up a spill ("I spilled my drink, can you help?"). A language model may respond with a reasonable narrative that is not feasible or useful for the robot. "You could try using a vacuum cleaner" is impossible if there is no vacuum in the scene or if the robot is incapable of using one. With prompt engineering, a LLM may be capable of splitting the high-level instruction into sub-tasks, but it cannot do so without the context of what the robot is capable of given its abilities *and* the current state of the robot *and* the environment.

Motivated by this example, we study the problem of how to extract the knowledge in LLMs for enabling an embodied agent, such as a robot, to follow high-level textual instructions. The robot is equipped with a repertoire of learned skills for "atomic" behaviors that are capable of low-level visuomotor control. We make use of the fact that, in addition to asking the LLM to simply interpret an instruction, we can use it to score the likelihood that an individual skill makes progress towards completing the high-level instruction. Then, if each skill has an affordance function that quantifies how likely it is to succeed from the current state (such as a learned value function), its value can be used to weight the skill's likelihood. In this way, the LLM describes the probability that each skill contributes to completing the instruction, and the affordance function describes the probability that each skill will succeed – combining the two provides the probability that each skill will perform the instruction *successfully*. The affordance functions make the LLM aware of the current scene, and constraining the completions to the skill descriptions makes the LLM aware of the robot's capabilities. Furthermore, this combination results in a fully explainable sequence of steps that the robot will execute to accomplish an instruction – an interpretable plan that is expressed through language.

Our method, SayCan, extracts and leverages the knowledge within LLMs in physically-grounded tasks. The LLM (Say) provides a task-grounding to determine useful actions for a high-level goal and the learned affordance functions (Can) provide a world-grounding to determine what is possible to execute upon the plan. We use reinforcement learning (RL) as a way to learn language-conditioned value functions that provide affordances of what is possible in the world. We evaluate the proposed approach on 101 real-world robotic tasks that involve a mobile robot accomplishing a large set of language instructions in a real kitchen in a zero-shot fashion. Our experiments validate that SayCan can execute temporally-extended, abstract instructions. Grounding the LLM in the real-world via affordances nearly doubles the performance over the non-grounded baselines.

## 2   SayCan: Do As I Can, Not As I Say

**Problem Statement**. Our system receives a user-provided natural language instruction $i$ that describes a task that the robot should execute. The instruction can be long, abstract, or ambiguous. We also assume that we are given a set of skills $\Pi$, where each skill is a short task, such as picking up a particular object, and comes with a short language description $\ell_\pi$ (e.g., "find a sponge") and an affordance function $p(c_\pi|s, \ell_\pi)$, which indicates the probability of $c$-completing the skill with description $\ell_\pi$ successfully from state $s$; where state can correspond to the robot, environment, or any information required to compute affordances. Intuitively, $p(c_\pi|s, \ell_\pi)$ means "if I ask the robot to do $\ell_\pi$, will it do it?". In RL terminology, $p(c_\pi|s, \ell_\pi)$ is the value function for the skill in an undiscounted setting if we take the reward to be 1 for successful completion and 0 otherwise.

As mentioned above, $\ell_\pi$ denotes the textual label of skill $\pi$ and $p(c_\pi|s, \ell_\pi)$ denotes the probability that skill $\pi$ with textual label $\ell_\pi$ successfully completes if executed from state $s$, where $c_\pi$ is a Bernoulli random variable. The LLM provides us with $p(\ell_\pi|i)$, the probability that a skill's textual label is a valid next step for the user's instruction. However, what we are interested in is the probability that a given skill successfully makes progress toward actually completing the instruction, which we denote as $p(c_i|i, s, \ell_\pi)$. Assuming that a skill that succeeds makes progress on $i$ with probability $p(\ell_\pi|i)$ (i.e., its probability of being the right skill), and a skill that fails makes progress with probability zero, we can factorize this as $p(c_i|i, s, \ell_\pi) \propto p(c_\pi|s, \ell_\pi)p(\ell_\pi|i)$. This corresponds to multiplying the probability of the language description of the skill given the instruction $p(\ell_\pi|i)$,

which we refer to as task-grounding, and the probability of the skill being possible in the current state of the world $p(c_\pi|s, \ell_\pi)$, which we refer to as world-grounding.

**Connecting Large Language Models to Robots.** While large language models can draw on a wealth of knowledge learned from copious amounts of text, they will not necessarily break down high-level commands into low-level instructions that are suitable for robotic execution. If a language model were asked "how would a robot bring me an apple", it may respond "a robot could go to a nearby store and purchase an apple for you". Though this response is a reasonable completion for the prompt, it is not necessarily actionable to an embodied agent, which may have a narrow and fixed set of abilities. Therefore, to adapt language models to our problem statement, we must somehow inform them that we specifically want the high-level instruction to be broken down into sequences of available low-level skills. One approach is careful prompt engineering [2, 3], a technique to coax a language model to a specific response structure. Prompt engineering provides examples in the context text ("prompt") for the model that specify the task and the response structure which the model will emulate; the prompt used in this work is shown in Appendix D.4 along with experiments ablating it. However, this is not enough to fully constrain the output to admissible primitive skills for an embodied agent, and indeed at times it can produce inadmissible actions or language that is not formatted in a way that is easy to parse into individual steps.

Scoring language models open an avenue to constrained responses by outputting the probabilities assigned by a language model to fixed outputs. A language model represents a *distribution* over potential completions $p(w_k|w_{<k})$, where $w_k$ is a word that appears at a $k^{\text{th}}$ position in a text. While typical generation applications (e.g., conversational agents) sample from this distribution or decode the maximum likelihood completion, we can also use the model to *score* a candidate completion selected from a set of options. Formally in SayCan, given a set of low-level skills $\Pi$, their language descriptions $\ell_\Pi$ and an instruction $i$, we compute the probability of a language description of a skill $\ell_\pi \in \ell_\Pi$ making progress towards executing the instruction $i$: $p(\ell_\pi|i)$, which corresponds to querying the model over potential completions. The optimal skill according to the language model is computed via $\ell_\pi = \arg\max_{\ell_\pi \in \ell_\Pi} p(\ell_\pi|i)$. Once selected, the process proceeds by iteratively selecting a skill and appending it to the instruction. Practically, in this work we structure the planning as a dialog between a user and a robot, in which a user provides the high level-instruction (e.g., "Bring me a coke can.") and the language model responds with an explicit sequence ("1. $\ell_\pi$", e.g., "1. find a coke can, 2. pick up the coke can, 3. bring it to you").

This has the added benefit of interpretability, as the model not only outputs generative responses, but also gives a notion of likelihood across many possible responses. Figure 2 (and Appendix Figure 9 in more detail) shows this process of forcing the LLM into a language pattern, where the set of tasks are the skills the low-level policy is capable of and prompt engineering shows plan examples and dialog between the user and the robot. With this approach, we are able to effectively extract knowledge from the language model, but it leaves a major issue: while the decoding of the instruction obtained in this way always consists of skills that are available to the robot, these skills may not necessarily be appropriate for executing the desired high-level task in the *specific* situation that the robot is currently in. For example, if I ask a robot to "bring me an apple", the optimal set of skills changes if there is no apple in view or if it already has one in its hand.

**SayCan.** The key idea of SayCan is to ground large language models through value functions – affordance functions that capture the log likelihood that a particular skill will be able to succeed in the current state. Given a skill $\pi \in \Pi$, its language description $\ell_\pi$ and its corresponding value function, which provides $p(c_\pi|s, \ell_\pi)$, the probability of $c$-ompletion for the skill described by $\ell_\pi$ in state $s$, we form an affordance space $\{p(c_\pi|s, \ell_\pi)\}_{\pi \in \Pi}$. This value function space captures affordances across all skills [4]. For each skill, the affordance function and the LLM probability are then multiplied together and ultimately the most probable skill is selected, i.e. $\pi = \arg\max_{\pi \in \Pi} p(c_\pi|s, \ell_\pi)p(\ell_\pi|i)$. Once the skill is selected, the corresponding policy is executed by the agent and the LLM query is amended to include $\ell_\pi$ and the process is run again until a termination token (e.g., "done") is chosen. This process is shown in Figure 2 and described in Appendix Algorithm 1. These two mirrored processes together lead to a probabilistic interpretation of SayCan, where the LLM provides probabilities of a skill being useful for the high-level instruction and the affordances provide probabilities of successfully executing each skill. Combining these two probabilities together provides a probability that this skill furthers the execution of the high-level instruction commanded by the user.

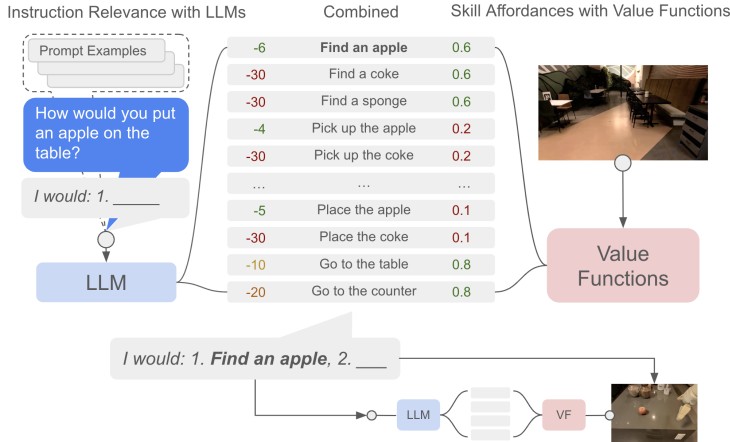

Figure 2: Given a high-level instruction, SayCan combines probabilities from a LLM (the probability that a skill is useful for the instruction) with the probabilities from a value function (the probability of successfully executing said skill) to select the skill to perform. This emits a skill that is both possible and useful. The process is repeated by appending the skill to the response and querying the models again, until the output step is to terminate. Appendix Figures 9 and 10 focus on the LLM and VFS components.

## 3 Implementing SayCan in a Robotic System

**Language-Conditioned Robotic Control Policies.** To instantiate SayCan, we must provide it with a set of skills, each of which has a policy, a value function, and a short language description (e.g., "pick up the can"). These skills, value functions, and descriptions can be obtained in a variety of different ways. In our implementation, we train the individual skills either with image-based behavioral cloning, following the BC-Z method [5], or reinforcement learning, following MT-Opt [6] and using features from PI-QT-Opt [7]. Regardless of how the skill's policy is obtained, we utilize value functions trained via TD backups as the affordance model for that skill. While we find that the BC policies achieve higher success rates at the current stage of our data collection process, the value functions provided by the RL policies are crucial as an abstraction to translate control capabilities to a semantic understanding of the scene. To amortize the cost of training many skills, we utilize multi-task BC and multi-task RL, where instead of training a separate policy and value function per skill, we train multi-task policies and models that are *conditioned* on the language description. Note, however, that this description only corresponds to *low level* skills – it is still the role of the LLM in SayCan to interpret the high-level instruction and break it up into individual low level skills.

To condition the policies on language, we utilize a pre-trained large sentence encoder language model [8]. We freeze the language model parameters during training and use the embeddings generated by passing in text descriptions of each skill. These text embeddings are used as the input to the policy and value function that specify which skill should be performed (see the details of the architectures used in the Appendix C.1). Since the language model used to generate the text embeddings is not necessarily the same as the language model used for planning, SayCan is able to utilize different language models well suited for different abstraction levels – understanding planning with respect to many skills as opposed to expressing specific skills more granularly.

**Training the Low-Level Skills.** We utilize both BC and RL policy training procedures to obtain the language-conditioned policies and value functions, respectively. To complete the description of the underlying MDP that we consider, we provide the reward function as well as the skill specification that is used by the policies and value functions. As mentioned previously, for skill specification we use a set of short, natural language descriptions that are represented as language model embeddings. We utilize sparse reward functions with reward values of $1.0$ at the end of an episode if the language command was executed successfully, and $0.0$ otherwise. An episode is marked successful if two out three human raters agree (given the instruction and video).

To learn language-conditioned BC policies at scale in the real world, we build on top of BC-Z [5] and use a similar policy-network architecture (shown in Fig. 7). To learn a language-conditioned RL policy, we use MT-Opt [6] with PI-QT-Opt features [7] in the Everyday Robots simulator using RetinaGAN sim-to-real transfer [9], implemented in TF-Agents [10]. We bootstrap the performance of simulation policies by utilizing simulation demonstrations to provide initial successes, and then continuously improve the RL performance with online data collection. We use a network architecture similar to MT-Opt (shown in Fig. 6). The action space of our policies includes the six degrees of freedom of the end-effector pose as well as gripper open and close commands, x-y position and yaw orientation delta of the mobile base of the robot, and the *terminate* action. Additional details on data collection and training are in Appendix Section C.2.

**Robotic System and Skills.** For the control policies, we study a diverse set of manipulation and navigation skills using a mobile manipulator robot. Inspired by common skills one might pose to a robot in a kitchen environment, we propose 551 skills that span seven skill families and 17 objects. The skill families are picking, placing, and rearranging objects, placing objects in specific configurations, opening and closing drawers, and navigating to various locations. A skill may then be "pick up the apple" or "go to the trash can". In this study we utilize the skills that are most amenable to more complex behaviors via composition and planning as well as those that have high performance at the current stage of data collection; for more details, see Appendix D.

# 4 Experimental Evaluation

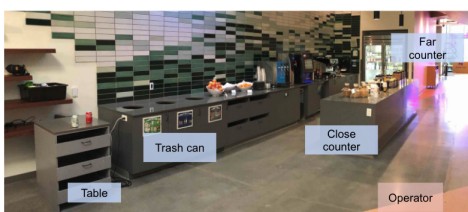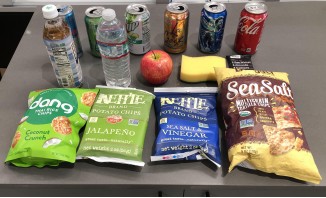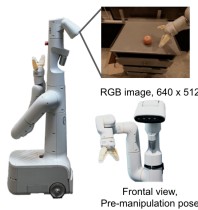

Figure 3: The experiments were performed in an office kitchen and a mock kitchen mirroring this setup, with 5 locations and 15 objects. The robot is a mobile manipulator with policies trained from an RGB observation.

**Experimental Setup.** We evaluate SayCan with a mobile manipulator and a set of object manipulation and navigation skills in two office kitchen environments. Figure 3 shows the environment setup and the robot. We use 15 objects commonly found in an office kitchen and 5 known locations with semantic meaning (two counters, a table, a trash can, and the user location). We test our method in two environments: a real office kitchen and a mock environment mirroring the kitchen, which is also the environment in which the robot's skills were trained. The mobile manipulator is from Everyday Robots and has a 7 DoF arm and a two-fingered gripper. The LLM used is 540B PaLM [11].

**Instructions.** To evaluate SayCan, we test across 101 instructions from 7 instruction families, summarized in Table 1 and enumerated in Appendix E.1, with the prompt shown in Listing 1. These were developed to test various aspects of SayCan and were inspired by crowd sourcing via Amazon Mechanical Turk and in-person kitchen users, as well as benchmarks such as ALFRED [12] and BE-HAVIOR [13]. The instructions, detailed in Table 1, span multiple axes of variation: time-horizon (from single primitives to 10+ in a row), language complexity (from structured language to fully crowd-sourced requests), and embodiment (variations over the robot and environment state).

| Instruction Family | Num | Explanation | Example Instruction |
|---|---|---|---|
| NL Single Primitive | 15 | NL queries for a single primitive | Let go of the coke can |
| NL Nouns | 15 | NL queries focused on abstract nouns | Bring me a fruit |
| NL Verbs | 15 | NL queries focused on abstract verbs | Restock the rice chips on the far counter |
| Structured Language | 15 | Structured language queries, mirror NL Verbs | Move the rice chips to the far counter. |
| Embodiment | 11 | Queries to test SayCan's understanding of the current state of the environment and robot | Put the coke on the counter. (starting from different completion stages) |
| Crowd-Sourced | 15 | Queries in unstructured formats | My favorite drink is redbull, bring one |
| Long-Horizon | 15 | Long-horizon queries that require many steps of reasoning | I spilled my coke on the table, throw it away and bring me something to clean |

Table 1: **List of instruction family definitions:** We evaluate the algorithm on 101 instructions. We group the instructions into different families, with each family focusing on testing one aspect of the proposed method.

**Metrics.** To understand the performance of SayCan we measure two main metrics. The first is **plan success rate**, which measures whether the skills selected by the model are correct for the instruction, regardless of whether or not they actually successfully executed. Note that for many instructions there may be multiple valid solutions. For example if the instruction is to "bring a sponge and throw away the soda can", the plan can choose to bring sponge first or throw away the soda can first. The second metric is **execution success rate**, which measures whether the full SayCan system actually performs the desired instruction successfully. For both, we ask three human raters to rate success.

## 4.1 Results

Table 2 shows the performance of SayCan across 101 tasks. In the mock kitchen, SayCan achieved a planning success rate of 84% and an execution rate of 74%. We also investigate SayCan out of the

lab setting and in the real kitchen to verify the performance of the policies and value functions in this setting. We find a reduction of planning performance by 3% and execution by 14%, indicating SayCan and the underlying policies generalize reasonably well to the full kitchen. The full task list and results can be found in the Appendix Table 5, LLM size ablations can be found in Appendix D.3, and videos of experiment rollouts and the decision making process can be found in the supplemental and on the project website: `say-can.github.io`.

Figure 4 shows two long-horizon queries and the resulting rollouts. These tasks require SayCan to plan many steps without error and for the robot to navigate and interact with a significant portion of the kitchen. Each query requires SayCan to understand context implicit within the instruction. In Figure 4a, the algorithm must understand the operator has asked for something to "recover from a workout", i.e. something healthy, and thus it brings water and an apple rather than, e.g., a soda and chips. Furthermore, the algorithm must understand ordering and history, that it has already brought a drink and now must bring a snack before terminating. In Figure 4b, SayCan must track which objects are the "them" that need to be disposed of and where the sponge should be brought.

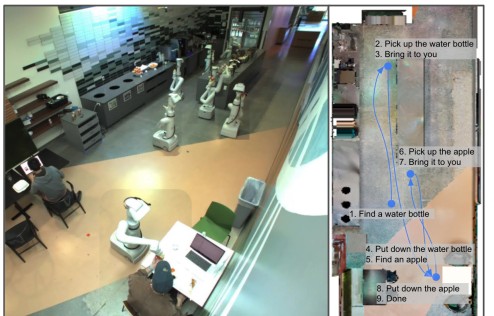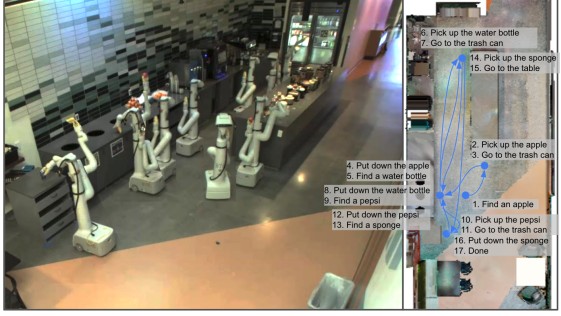

(a) "I just worked out, can you bring me a drink and a snack to recover?"

(b) "I left out a coke, apple, and water, can you throw them away and then bring me a sponge to wipe the table?"

Figure 4: Timelapse of rollouts to two long-horizon queries. The robot interacts with a large portion of the kitchen environment and successfully performs sequences of manipulation and navigation skills.

Figure 5 highlights SayCan's decision making, along with its interpretability. The decision making process can be understood as it solves instructions by visualizing what the two sides of the algorithm output. This allows a user to understand what options SayCan is considering as language completions and what it believes is possible. We find that sequence order is understood (approaching objects before picking them up and picking them up before bringing them). Figure 5 shows that though the query mentions a coke, SayCan understands that the important object is something to clean and brings a sponge. Appendix E.5 shows additional rollouts with complex decisions, embodiment grounding, and long-horizon tasks in Figures 12, 13, 15 as well as failures in Figure 14. We believe such real-time and clear interpretability opens avenues to more interactive operation.

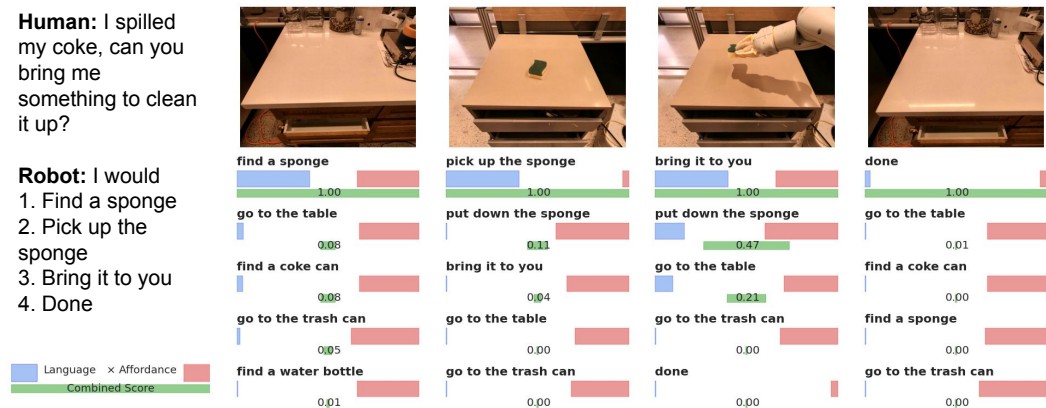

Figure 5: Visualization of SayCan's decision making, where the top combined score chooses the correct skill.

When comparing the performance of different instruction families in Table 2 (see Table 1 for an explanation of families), we see that the natural language nouns performed worse than natural lan-

guage verbs, due to the number of nouns possible (15 objects and 5 locations) versus number of verbs (6). The structured language tasks (created to ablate the performance loss of spelling out the solution versus understanding the query) were planned correctly 93% of the time, while their natural language verb counterparts were planned correctly 100%. This indicates the language model effectively parses the queries. The embodiment tasks were planned correctly 64% of the time, generally with failures as a result of affordance function misclassification. SayCan planned and executed crowd-sourced natural queries with performance on par with other instruction families. SayCan performed worst on the most challenging long-horizon tasks, where most failures were a result of early termination by the LLM (e.g., bringing one object but not the second). We also find that Say-Can struggles with negation (e.g., "bring me a snack that isn't an apple") and ambiguous references (e.g. asking for drinks with caffeine), which is a known issue inherited from underlying language models [14]. Overall, 65% of the errors were LLM failures and 35% were affordance failures.

**Ablating Language.** To study the importance of the LLM, we conduct two ablation experiments using the language-conditioned policy (see Sections 3-3). In *BC NL* we feed the full instruction $i$ into the policy – this approach is representative of standard RL or BC-based instruction following methods [5, 15, 16, 17]. In *BC USE* we project the high-level instruction into the set of known language commands via the Universal Sentence Encoder (USE) embeddings [8] by embedding the instruction, all the tasks, and the combinatorial set of sequences tasks (i.e., we consider "pick coke can" as well as "1. find coke can, 2. pick coke can" and so on), and selecting the highest cosine similarity instruction. The results in Table 2 illustrate the necessity of the language grounding where *BC NL* achieves 0% in all tasks and *BC USE* achieves 60% for single primitives, but 0% otherwise.

**Ablating Value Functions.** Table 2 illustrates the necessity of the affordance grounding. We compare SayCan to (1) *No VF*, which removes the value function grounding (i.e., choosing the maximum language score skill) and to (2) *Generative*, which uses the generative output of the LLM and then projects each planned skill to its maximal cosine similarity skill via USE embeddings. The latter in effect compares to [18], which loses the explicit option probabilities, and thus is less interpretable and cannot be combined with affordance probabilities. For *Generative* we also tried BERT embeddings [19], but found poor performance. The *No VF* and *Generative* approaches performed similarly, achieving 67% and 74% planning success rate respectively, and worse than SayCan's 84%.

| | | Mock Kitchen | | Kitchen | | No Affordance | | No LLM | |
|---|---|---|---|---|---|---|---|---|---|
| | | **SayCan** | **SayCan** | **SayCan** | **SayCan** | **No VF** | **Gen.** | **BC NL** | **BC USE** |
| **Family** | **Num** | Plan | Execute | Plan | Execute | Plan | Plan | Execute | Execute |
| NL Single | 15 | 100% | 100% | 93% | 87% | 73% | 87% | 0% | 60% |
| NL Nouns | 15 | 67% | 47% | 60% | 40% | 53% | 53% | 0% | 0% |
| NL Verbs | 15 | 100% | 93% | 93% | 73% | 87% | 93% | 0% | 0% |
| Structured | 15 | 93% | 87% | 93% | 47% | 93% | 100% | 0% | 0% |
| Embodiment | 11 | 64% | 55% | 64% | 55% | 18% | 36% | 0% | 0% |
| Crowd Sourced | 15 | 87% | 87% | 73% | 60% | 67% | 80% | 0% | 0% |
| Long-Horizon | 15 | 73% | 47% | 73% | 47% | 67% | 60% | 0% | 0% |
| Total | 101 | 84% | 74% | 81% | 60% | 67% | 74% | 0% | 9% |

Table 2: Success rates of instructions by family. SayCan achieves a planning success rate of 84% and execution success rate of 74% in the training environment and 81% planning and 60% execution in a real kitchen. *No VF* uses the maximum score skill from the LLM, *Generative (Gen.)* uses a generative LLM and then projects to the nearest skill via USE embeddings, *BC NL* uses the policy with the natural language instruction, and *BC USE* uses the policy with the natural language instruction projected to the nearest skill via USE embeddings.

### 4.2 Case Study of New Capabilities

**Adding Skills: Drawer Manipulation (Appendix E.2).** SayCan is capable of more varied and complex sequences as well as integrating new skills by simply adding the new skills as options for the LLM and provide accompanying value functions and add an example in the prompt with that skill. For example, with the skills open, close, and go to the drawer, SayCan is capable of solving tasks such as "restock the coke and pepsi into the drawer", or the query shown in Appendix Figure 15. Over 21 queries we found a planning rate of 100% and an execution rate of 33% (due to failures of the chained manipulation policy), with no loss in performance for other instructions.

**Chain of Thought Reasoning (Appendix E.3).** SayCan can be integrated with recent work improving LLM reasoning, such as Chain of Thought [20].

**Multilingual Queries (Appendix E.4).** The LLM was trained on a multilingual corpus and thus SayCan can handle multilingual queries with comparable success rate as English.

# 5 Related Work

**Grounding Language Models.** A significant body of work has focused on grounding language [21, 22]. Recent works have studied how to ground modern language models, by training them to accept additional environment inputs [23, 24, 25, 26, 27, 28, 29] or to directly output actions [30, 31, 32]. Others grounded language in an environment through prompt engineering [20]. Concurrently with SayCan, Huang et al. [18] use prompt engineering to extract temporally extended plans, but without any additional mechanism to ensure grounding, roughly corresponding to the "Generative" baseline in our experiments. The above methods are all trained without interaction with a physical environment, thus limiting their ability to reason over embodied interactions. One approach to grounding language models in interaction is by learning downstream networks with pre-trained LLM representations [33, 17, 16, 34, 35, 36, 37, 38]. Another approach finetunes language models with interactive data, such as rewards or ranking feedback of the interaction [3, 39, 40]. In our work, SayCan grounds language models in the environment through previously-trained value functions, enabling general, long-horizon behaviors in a zero-shot manner, i.e., without additional training.

**Learning Language-Conditioned Behavior.** There is a long history of research studying how to connect language and behavior [41, 42, 43, 44, 45, 46]. A large number of prior works have learned language-conditioned behavior via imitation learning [47, 17, 15, 5, 48, 32] or reinforcement learning [49, 50, 44, 51, 52, 53, 16, 36]. Most focus on following low-level instructions [15, 17, 53, 16, 5, 48], though some methods address long-horizon, compound tasks in simulated domains [54, 55, 51]. Like these latter works, we focus on completing temporally extended tasks. However, a central aspect of our work is to solve such tasks by extracting and leveraging the knowledge in large language models. This enables a more flexible framework and more general abstract, tasks than semantic parsing [43, 46, 56]. While prior works have studied how pre-trained language embeddings can improve generalization to new instructions [33, 17, 16] and to new low-level tasks [5], we extract much more substantial knowledge from LLMs by grounding them within the robot's affordances.

**Task Planning and Motion Planning.** Task and motion planning [57, 58] is a problem of sequencing tasks to solve a high-level problem, while ensuring the feasibility given an embodiment (task planning [59, 60, 61]; motion planning [62]). Classically, this problem has been solved through symbolic planning [59, 61] or optimization [63, 64], but these require explicit primitives and constraints. Machine learning has recently been applied to enable abstract task specification, allow general primitives, or relax constraints [65, 66, 67, 68, 69, 70, 71, 72, 73, 74, 75, 76]. Others learn to hierarchically solve such long-horizon problems [77, 78, 4, 79, 51]. SayCan leverages an LLM's semantic knowledge about the world for interpreting instructions *and* understanding how to execute them. The use of LLMs and generality of learned low-level policies enables long-horizon, abstract tasks that scale effectively to the real world, as demonstrated in our robot experiments.

# 6 Conclusions and Limitations

We presented SayCan, a method that enables leveraging and grounding the rich knowledge in large language models to complete embodied tasks. For real-world grounding, we use RL-based value functions for the individual skills that provide affordances of what is possible in the world, and use textual labels for these skills as potential responses that are scored by a language model. We evaluated this approach on a number of real-world robotic tasks that involve a mobile manipulator robot accomplishing a large set of long-horizon natural language instructions in a real kitchen.

While SayCan presents a viable way to ground language models in agents' affordances, it has a number of limitations. First, we expect this method to inherit the limitations and biases of LLMs [80, 81], including the dependence on the training data. In addition, we observe that even though SayCan allows the users to interact with the agents using natural language commands, the primary bottleneck of the system is in the range and capabilities of the underlying skills. Beyond these skills, SayCan relies on reliable affordances to gauge the ability of the robot to perform a task given the state. Though with RL trained policies, the value function provides a reasonable approximation of the affordance, it is an open question how to best train affordances for general skills. Finally, a limitation of the proposed framework is that the LLM only receives feedback from the environment through the selected skills, which can be limited, e.g., if a skill fails. It would be useful to investigate to what degree sources of environment feedback can be incorporated, such as through success detectors, scene descriptions, direct visual feedback, or visual-language models. To illustrate limitations, we present failure cases in Appendix E and open sourced code in a tabletop robot setting in Appendix F.

## Acknowledgments

The authors would like to thank Yunfei Bai, Matt Bennice, Maarten Bosma, Justin Boyd, Bill Byrne, Kendra Byrne, Noah Constant, Pete Florence, Laura Graesser, Rico Jonschkowski, Daniel Kappler, Hugo Larochelle, Benjamin Lee, Adrian Li, Suraj Nair, Krista Reymann, Jeff Seto, Dhruv Shah, Ian Storz, Razvan Surdulescu, Tom Small, and Vincent Zhao for their help and support in various aspects of the project.

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

# A Contributions

## A.1 By Type

- **Designed and built distributed robot learning infrastructure:** Michael Ahn, Anthony Brohan, Noah Brown, Yevgen Chebotar, Byron David, Keerthana Gopalakrishnan, Karol Hausman, Alex Herzog, Daniel Ho, Jasmine Hsu, Julian Ibarz, Alex Irpan, Eric Jang, Nikhil J Joshi, Ryan Julian, Dmitry Kalashnikov, Yuheng Kuang, Yao Lu, Peter Pastor, Kanishka Rao, Nicolas Sievers, Fei Xia, Ted Xiao, Peng Xu, Sichun Xu, and Mengyuan Yan.

- **Designed, implemented, or trained the underlying manipulation policies:** Yevgen Chebotar, Keerthana Gopalakrishnan, Karol Hausman, Julian Ibarz, Alex Irpan, Eric Jang, Nikhil Joshi, Ryan Julian, Kuang-Huei Lee, Yao Lu, Kanishka Rao, and Ted Xiao.

- **Designed or implemented the data generation and curation or collected data:** Noah Brown, Omar Cortes, Jasmine Hsu, Alex Irpan, Eric Jang, Rosario Jauregui Ruano, Kyle Jeffrey, Linda Luu, Jornell Quiambao, Kanishka Rao, Jarek Rettinghouse, Diego Reyes, Pierre Sermanet, Clayton Tan, and Sichun Xu.

- **Designed or implemented SayCan:** Karol Hausman, Brian Ichter, Sergey Levine, Alexander Toshev, and Fei Xia.

- **Managed or advised on the project:** Chelsea Finn, Karol Hausman, Eric Jang, Sally Jesmonth, Sergey Levine, Yao Lu, Carolina Parada, Kanishka Rao, Alexander Toshev, and Vincent Vanhoucke.

- **Ran evaluations or experiments:** Noah Brown, Omar Cortes, Brian Ichter, Rosario Jauregui Ruano, Kyle Jeffrey, Linda Luu, Jornell Quiambao, Jarek Rettinghouse, Diego Reyes, Clayton Tan, and Fei Xia.

- **Scaled simulation infrastructure:** Nikhil J Joshi, Yao Lu, Kanishka Rao, and Ted Xiao.

- **Wrote the paper:** Chelsea Finn, Karol Hausman, Brian Ichter, Alex Irpan, Sergey Levine, Fei Xia, and Ted Xiao.

## A.2 By Person

**Michael Ahn** developed the deployment system that enabled the ability to scale up data collection on real robots.

**Anthony Brohan** implemented the logging system for the project and designed and implemented the data labeling pipelines.

**Noah Brown** led and coordinated the real-robot operations including data collection with teleoperators, evaluations and the real-world setup.

**Yevgen Chebotar** designed and implemented multiple offline RL methods allowing the manipulation policies to process data coming from different sources.

**Omar Cortes** collected data on the robots and ran and supervised real-world evaluations.

**Byron David** developed simulation assets and performed system identification.

**Chelsea Finn** advised on the project, helped set the research direction and wrote parts of the paper.

**Keerthana Gopalakrishnan** provided multiple infrastructure contributions that allowed for scalable learning of manipulation policies.

**Karol Hausman** co-led the project as well as developed SayCan, helped set the research direction, trained the underlying manipulation policies, and wrote the paper.

**Alex Herzog** developed the teleoperation tools and implemented multiple infrastructure tools that allowed for continuous robot operation.

**Daniel Ho** helped develop sim-to-real pipelines for manipulation policies.

**Jasmine Hsu** provided logging and monitoring infrastructure tools as well as data labeling pipelines.

**Julian Ibarz** provided multiple contributions that enabled scaling learning algorithms for manipulation policies, and helped set the research direction.

**Brian Ichter** initiated and led the SayCan algorithm, combined the manipulation and navigation skills, ran experiments for the paper, and wrote the paper.

**Alex Irpan** set up and led the autonomous data collection effort as well as verified the data collected by the robots, and wrote parts of the paper.

**Eric Jang** helped set the research and team direction, managed the data for learning, developed the behavioral cloning manipulation policies, and wrote parts of the paper.

**Rosario Jauregui** Ruano collected data on the robots and ran and supervised real-world evaluations.

**Kyle Jeffrey** collected data on the robots and ran and supervised real-world evaluations.

**Sally Jesmonth** was the program manager for the project.

**Nikhil J Joshi** developed a number of simulation and infrastructure tools that allowed to scale up simulation training.

**Ryan Julian** developed multi-modal network architectures and trained manipulation policies.

**Dmitry Kalashnikov** contributed infrastructure pieces that enabled training from logged data.

**Yuheng Kuang** implemented the logging system for the project and designed and implemented the data labeling pipelines

**Kuang-Huei Lee** made improvements to training algorithms for manipulation policies.

**Sergey Levine** advised on the project, helped set the research direction, developed SayCan, and wrote parts of the paper.

**Yao Lu** led and designed the robot learning infrastructure for the project providing most of the tools and improving manipulation policies.

**Linda Luu** ran multiple evaluations, collected data and helped establish real-robot operations.

**Carolina Parada** advised on the project, managed the team, helped write the paper, and helped set the research direction.

**Peter Pastor** provided infrastructure tools that allowed for continuous robot operations.

**Jornell Quiambao** collected data on the robots and ran and supervised real-world evaluations.

**Kanishka Rao** co-led the project, managed the team, helped set the research direction and contributed to training manipulation policies.

**Jarek Rettinghouse** collected data on the robots and ran and supervised real-world evaluations.

**Diego Reyes** collected data on the robots and ran and supervised real-world evaluations.

**Pierre Sermanet** set up the crowd compute rating pipeline.

**Nicolas Sievers** provided simulation assets and environments used for simulation training.

**Clayton Tan** collected data on the robots and ran and supervised real-world evaluations and helped establish real-robot operations.

**Alexander Toshev** advised on the project, developed SayCan, helped write the paper, and helped set research direction.

**Vincent Vanhoucke** advised on the project, managed the team, and helped write the paper.

**Fei Xia** developed, implemented, and led on-robot SayCan, ran the experiments for the paper, created the demos, and wrote the paper.

**Ted Xiao** led the scaling of manipulation skills, designed and developed learning from simulation for manipulation skills, and developed multi-modal network architectures.

**Peng Xu** was the engineering lead for integrating manipulation and navigation and developed the underlying infrastructure for SayCan.

**Sichun Xu** developed remote teleoperation tools that allowed scaling up data collection in simulation.

**Mengyuan Yan** implemented infrastructure and learning tools that allowed for learning manipulation policies from different data sources.

### A.3 Corresponding Emails:

{ichter,xiafei,karolhausman}@google.com

## B Background

**Large Language Models.** Language models seek to model the probability $p(W)$ of a text $W = \{w_0, w_1, w_2, ..., w_n\}$, a sequence of strings $w$. This is generally done through factorizing the probability via the chain rule to be $p(W) = \Pi_{j=0}^n p(w_j|w_{<j})$, such that each successive string is predicted from the previous. Recent breakthroughs initiated by neural network-based Attention architectures [82] have enabled efficient scaling of so-called Large Language Models (LLMs). Such models include Transformers [82], BERT [19], T5 [83], GPT-3 [2], Gopher [84], LAMDA [85], FLAN [86], and PaLM [11], each showing increasingly large capacity (billions of parameters and terabytes of text) and subsequent ability to generalize across tasks.

In this work, we utilize the vast semantic knowledge contained in LLMs to determine useful tasks for solving high-level instructions.

**Value functions and RL.** Our goal is to be able to accurately predict whether a skill (given by a language command) is feasible at a current state. We use temporal-difference-based (TD) reinforce-

ment learning to accomplish this goal. In particular, we define a Markov decision process (MDP) $\mathcal{M} = (\mathcal{S}, \mathcal{A}, P, R, \gamma)$, where $\mathcal{S}$ and $\mathcal{A}$ are state and action spaces, $P : \mathcal{S} \times \mathcal{A} \times \mathcal{S} \rightarrow \mathbb{R}_+$ is a state-transition probability function, $R : \mathcal{S} \times \mathcal{A} \rightarrow \mathbb{R}$ is a reward function and $\gamma$ is a discount factor. The goal of TD methods is to learn state or state-action value functions (Q-function) $Q^\pi(s, a)$, which represents the discounted sum of rewards when starting from state $s$ and action $a$, followed by the actions produced by the policy $\pi$, i.e. $Q^\pi(s, a) = \mathbb{E}_{a \sim \pi(a|s)} \sum_t R(s_t, a_t)$. The Q-function, $Q^\pi(s, a)$ can be learned via approximate dynamic programming approaches that optimize the following loss: $L_{TD}(\theta) = \mathbb{E}_{(s,a,s') \sim \mathcal{D}} [R(s, a) + \gamma \mathbb{E}_{a^* \sim \pi} Q^\pi_\theta(s', a^*) - Q^\pi_\theta(s, a)]$, where $\mathcal{D}$ is the dataset of states and actions and $\theta$ are the parameters of the Q-function.

In this work, we utilize TD-based methods to learn said value function that is additionally conditioned on the language command and utilize those to determine whether a given command is feasible from the given state. It is worth noting that in the sparse reward case, where the agent receives the reward of 1.0 at the end of the episode if it was successful and 0.0 otherwise, the value function trained via RL corresponds to an affordance function [87] that specifies whether a skill is possible in a given state. We leverage that intuition in our setup and express affordances via value functions of sparse reward tasks.

## C    RL and BC Policies

### C.1    RL and BC Policy Architecture

The RL models use an architecture similar to MT-Opt [6], with slight changes to support natural language inputs (see Fig. 6 for the network diagram). The camera image is first processed by 7 convolutional layers. The language instruction is embedded by the LLM, then concatenated with the robot action and non-image parts of the state, such as the gripper height. To support asynchronous control, inference occurs while the robot is still moving from the previous action. The model is given how much of the previous action is left to execute [88]. The conditioning input goes through FC layers, then tiled spatially and added to the conv. volume, before going through 11 more convolutional layers. The output is gated through a sigmoid, so the Q-value is always in $[0, 1]$.

The BC models use an architecture similar to BC-Z [5] (see Fig. 7 for the network diagram). The language instruction is embedded by a universal sentence encoder [8], then used to FiLM condition a Resnet-18 based architecture. Unlike the RL model, we do not provide the previous action or gripper height, since this was not necessary to learn the policy. Multiple FC layers are applied to the final visual features, to output each action component (arm position, arm orientation, gripper, and the termination action).

### C.2    RL and BC Policy Training

**RL training.** In addition to using demonstrations in the BC setup, we also learn language-conditioned value functions with RL. For this purpose, we complement our real robot fleet with a simulated version of the skills and environment. To reduce the simulation-to-real gap we transform robot images via RetinaGAN [9] to look more realistic while preserving genera object structure. In order to learn a language-conditioned RL policy, we utilize MT-Opt [6] in the Everyday Robots simulator using said simulation-to-real transfer. We bootstrap the performance of simulation policies by utilizing simulation demonstrations to provide initial successes, and then continuously improve the RL performance with online data collection in simulation. Standard image augmentations (random brightness and contrast) as well as random cropping were applied. The 640 x 512 input image was padded by 100 pixels left-right and 40 pixels top-down, then cropped back down to a 640 x 512 image, so as to allow for random spatial shifts without limiting the field of view. We use a network architecture similar to MT-Opt (shown in Fig. 6).

The RL model is trained using 16 TPUv3 chips and for about 100 hours, as well as a pool of 3000 CPU workers to collect episodes and another 3000 CPU workers to compute target Q-values. Computing target Q-values outside the TPU allows the TPU to be used solely for computing gradient updates. Episode rewards are sparse and always 0 or 1, so the Q-function is updated using a log loss. Models were trained using prioritized experience replay [89], where episode priority was tuned to encourage replay buffer training data for each skill to be close to 50% success. Episodes were sampled proportionally to their priority, defined as $1 + 10 \cdot |p - 0.5|$, where $p$ is the average success rate of episodes in the replay buffer.

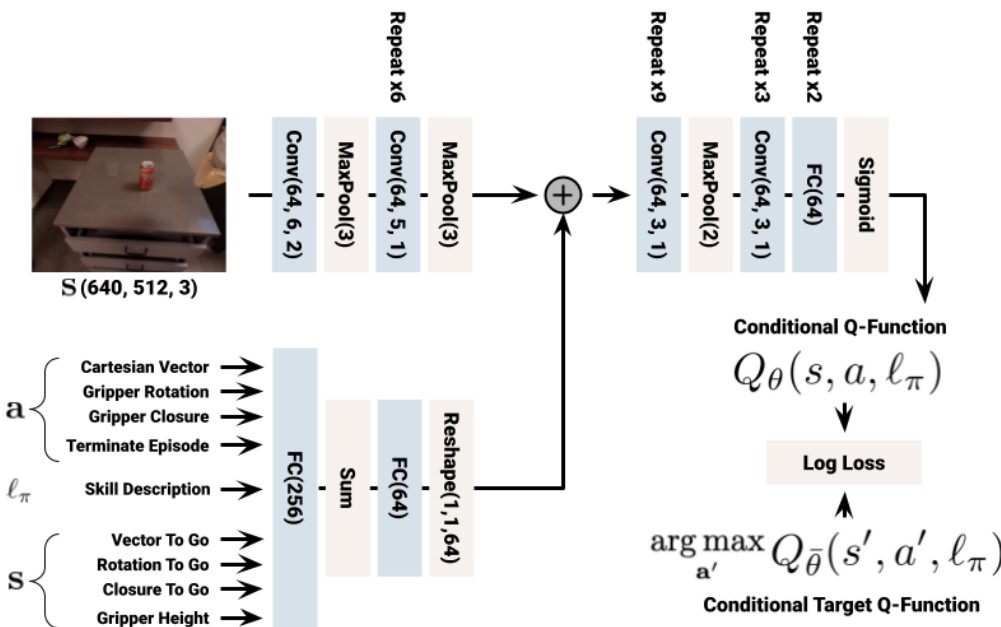

Figure 6: Network architecture in RL policy

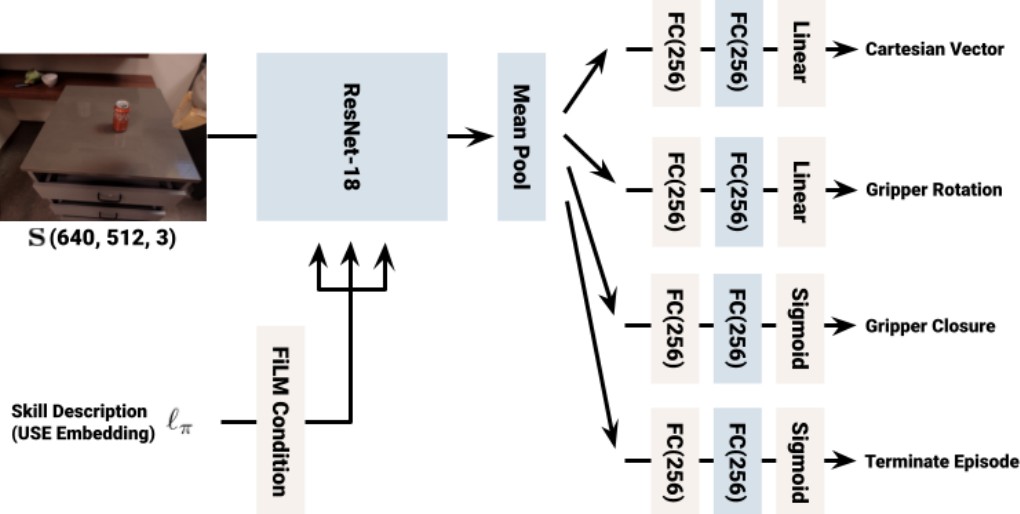

Figure 7: Network architecture in BC policy

**BC training.** We use 68000 teleoperated demonstrations that were collected over the course of 11 months using a fleet of 10 robots. The operators use VR headset controllers to track the motion of their hand, which is then mapped onto the robot's end-effector pose. The operators can also use a joystick to move the robot's base. We expand the demonstration dataset with 276000 autonomous episodes of learned policies which are later success-filtered and included in BC training, resulting in an additional 12000 successful episodes. To additionally process the data, we also ask the raters to mark the episodes as unsafe (i.e., if the robot collided with the environment), undesirable (i.e., if the robot perturbed objects that were not relevant to the skill) or infeasible (i.e., if the skill cannot be done or is already accomplished). If any of these conditions are met, the episode is excluded from training.

To learn language-conditioned BC policies at scale in the real world, we build on top of BC-Z [5] and use a similar policy-network architecture (shown in Fig. 7). It is trained with an MSE loss

for the continuous action components, and a cross-entropy loss for the discrete action components. Each action component was weighted evenly. Standard image augmentations (random brightness and contrast) as well as random cropping were used. The 640 x 512 input image was padded by 100 pixels left-right and 40 pixels top-down, then cropped back down to a 640 x 512 image, so as to allow for random spatial shifts without limiting the field of view. For faster iteration speeds with negligible training performance reduction, image inputs were down sampled to half-size (256 x 320 images). Affordance value functions were trained with full-size images, since half-size images did not work as well when learning $Q(s, a, \ell_\pi)$. The BC model is trained using 16 TPUv3 chips and trained for about 27 hours.

### C.3 RL and BC Policy Evaluations

In order to obtain the best possible manipulation capabilities for use in SayCan, we use a separate evaluation protocol for iterating on the RL and BC policies in the Mock Office Kitchen stations. Evaluations are divided by skill (pick up, knock over, place upright, open/close drawers, move object close to another one), and within each skill, 18-48 skills are sampled from a predetermined set of three objects. Object positions are randomized on each episode, with one or two objects serving as a distractor.

The episode ends when 50 actions have been taken or the policy samples a terminate action. A human operator supervises multiple robots performing evaluation and performs scene resets as needed, and records each episode as a success or failure. Models whose per-skill performance outperforms prior models are "graduated" to the same evaluation protocol in the real kitchen, and then integrated into SayCan. We found that despite the domain shift from Mock Office Kitchen stations to the actual kitchen counter and drawers, higher success rates on mock stations usually corresponded to higher success rates in the real kitchen setting.

Figure 8 shows the development of the manipulation skills over time. It reports the per-skill success rate, the average success rate across all skills, and the number of instructions the policy was trained on. Over the course of the project, we increased the number of skills evaluated, from 1 instruction in April 2021 to hundreds of instructions at time of publication over the course of 366 real-world model evaluations.

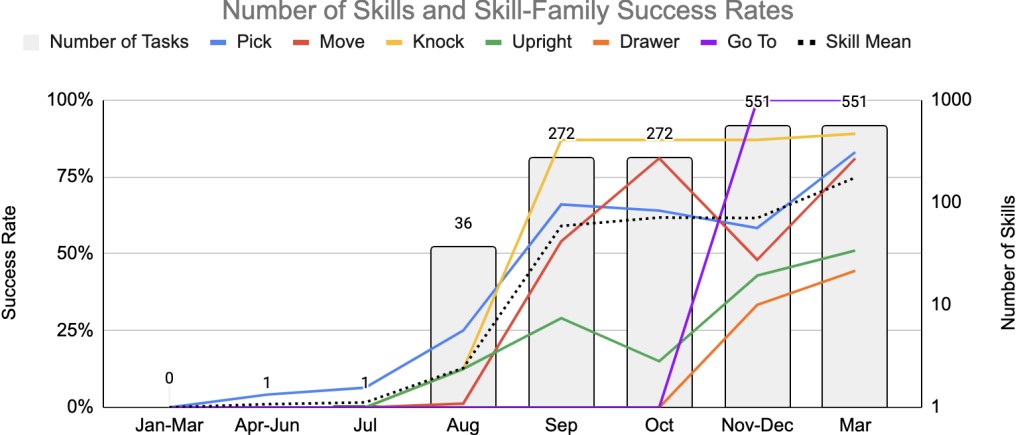

Figure 8: Per-skill evaluation performance of the best policies and number of skills over the duration of the project. The performance as well as the number of skills that the robots are able to handle grow over time due to the continuous data collection efforts as well as improving the policy training algorithms.

## D SayCan Details and Parameters

### D.1 SayCan Details

Figure 9 shows scoring approach used and prompt engineering for the LLM side of SayCan. Figure 10 shows how robotic affordances are computed with value functions and real value function computations at different states. These two components are combined to form SayCan, as detailed in Algorithm 1 and in Figure 11.

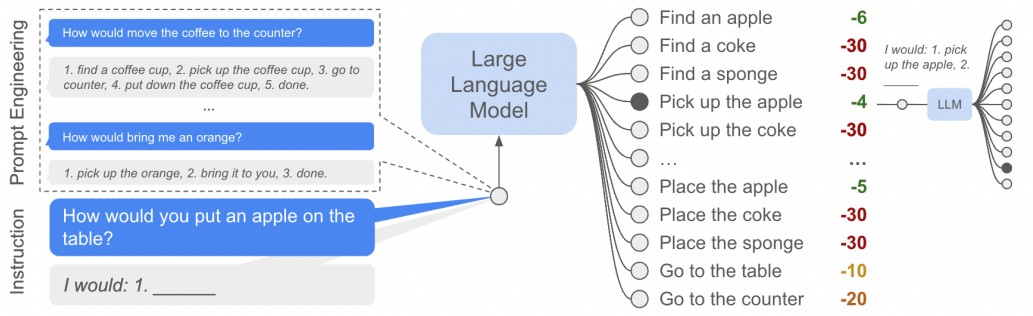

Figure 9: A scoring language model is queried with a prompt-engineered context of examples and the high-level instruction to execute and outputs the probability of each skill being selected. To iteratively plan the next steps, the selected skill is added to the natural language query and the language model is queried again.

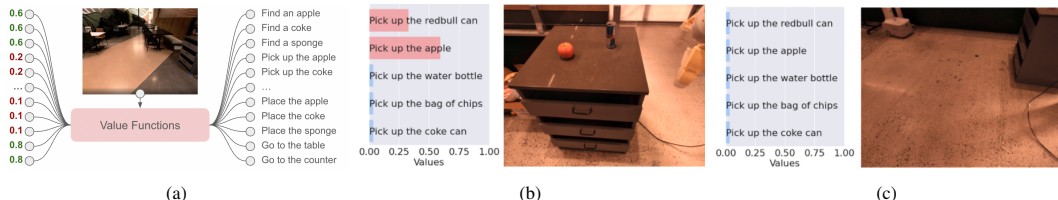

(a)                              (b)                              (c)

Figure 10: A value function module (a) is queried to form a value function space of action primitives based on the current observation. Visualizing "pick" value functions, in (b) "Pick up the red bull can" and "Pick up the apple" have high values because both objects are in the scene, while in (c) the robot is navigating an empty space, and thus none of the pick up actions receive high values.

## D.2 Skills, Policies, and Affordance Functions

We also note a few practical considerations for setting up our affordance functions and policies. The flexibility of our approach allows us to mix and match policies and affordances from different methods. For the pick manipulation skills we use a single multi-task, language-conditioned policy, for the place manipulation skills we use a scripted policy with an affordance based on the gripper state, and for navigation policies we use a planning-based approach which is aware of the locations where specific objects can be found and a distance measure. In order to avoid a situation where a skill is chosen but has already been performed or will have no effect, we set a cap for the affordances indicating that the skill has been completed and the reward received.

SayCan is capable of incorporating many different policies and affordance functions through its probability interface. Though in principle each type of skill has been trained with the pipeline described in Appendix C, to the success rates seen in Figure 8, we wish to show the generality of SayCan to different policies and affordance functions as well as the robustness of other functions (e.g. distance for navigation). Furthermore, some skills (such as the manipulation skill "move

---

**Algorithm 1** SayCan

**Given:** A high level instruction $i$, state $s_0$, and a set of skills $\Pi$ and their language descriptions $\ell_\Pi$

1:  $n = 0, \pi = \emptyset$
2:  **while** $\ell_{\pi_{n-1}} \neq$ "done" **do**
3:      $\mathcal{C} = \emptyset$
4:      **for** $\pi \in \Pi$ and $\ell_\pi \in \ell_\Pi$ **do**
5:          $p_\pi^{\text{LLM}} = p(\ell_\pi | i, \ell_{\pi_{n-1}}, ..., \ell_{\pi_0})$                    ▷ Evaluate scoring of LLM
6:          $p_\pi^{\text{affordance}} = p(c_\pi | s_n, \ell_\pi)$                    ▷ Evaluate affordance function
7:          $p_\pi^{\text{combined}} = p_\pi^{\text{affordance}} p_\pi^{\text{LLM}}$
8:          $\mathcal{C} = \mathcal{C} \cup p_\pi^{\text{combined}}$
9:      **end for**
10:     $\pi_n = \arg\max_{\pi \in \Pi} \mathcal{C}$
11:     Execute $\pi_n(s_n)$ in the environment, updating state $s_{n+1}$
12:     $n = n + 1$
13: **end while**

---

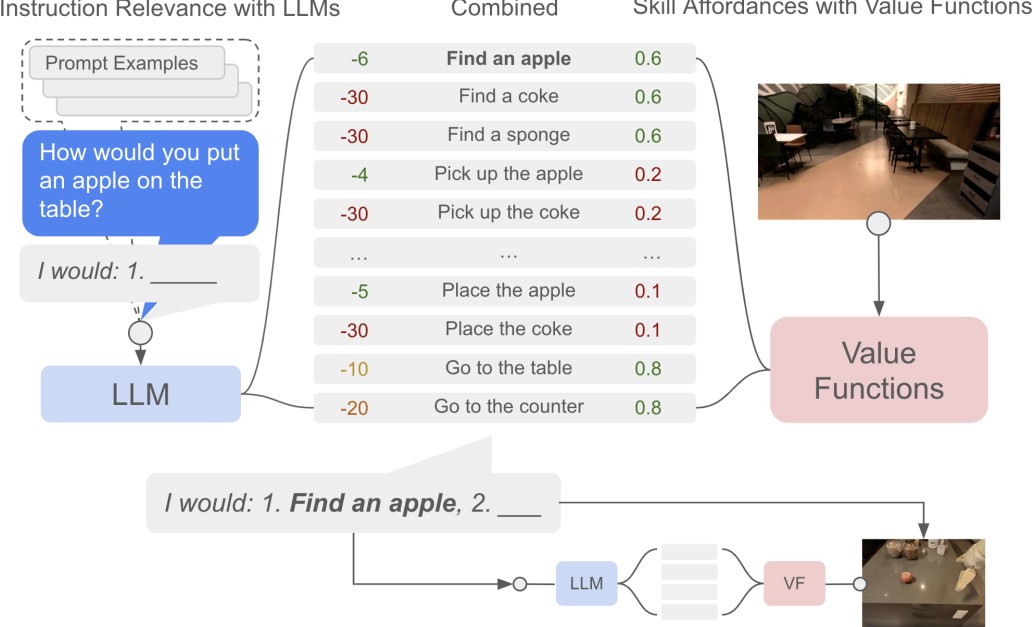

Figure 11: (A copy of Figure 2 here for clarity) Given a high-level instruction, SayCan combines probabilities from a language model (representing the probability that a skill is useful for the instruction) with the probabilities from a value function (representing the probability of successfully executing said skill) to select the skill to perform. This emits a skill that is both possible and useful. The process is repeated by appending the selected skill to the robot response and querying the models again, until the output step is to terminate.

object near object" and "knock object over") are not naturally part of long-horizon tasks and thus we do not utilize them. Other skills, such as drawer opening, were not consistent enough for long-horizon planning and thus unused. However, we note that as skills become performant or as new skills are learned, it is straightforward to incorporate these skills by adding them as options for LLM scoring and as examples in the prompt. We use the following for each skill family:

- **Pick.** For pick we use the learned policies in Appendix C and Section 3 with actions from BC and value functions from RL trained on the same skill. In natural language these are specified as "pick up the object".

- **Go to.** Since the focus of this work is mainly on planning, we assume the location of objects are known. Thus any navigation skill maps to the coordinate of the object with a classical planning-based navigation stack. In natural language these are specified as "go to location" and "find object".

- **Place.** Though our manipulation policies have a "place upright" skill, this skill only applies to objects that have a canonical upright direction, e.g., a water bottle but not a bag of chips. One could also train a universal "place" command, but our current policies are trained in a setup-free environment and thus are not amenable to an initial pick. Thus to have a consistent place policy across all objects we use a classical motion planning policy. We use Cartesian space motion planning to plan a path from pre-grasp pose shown in Figure 3 to a gripper release pose. The robot executes that path until the gripper is in contact with a supporting surface, and then the gripper opens and releases the object. In natural language these are specified as "put down the object".

Each skill is thus an explicit text command performed by a low-level policy for that skill. For object above, we use the objects shown in Figure 3 placed in random configurations at locations throughout the scene (e.g., coke can, water bottle, jalapeno chips, apple, sponge, etc.). These objects can be placed in random configurations at locations throughout the scene. For location above, we consider several named locations with known positions shown in Figure 3 (e.g., table, trash can, etc.). This results in skills such as the following:

- "Pick up the coke", "pick up the 7up", "pick up the apple", "pick up the sponge", etc. Note that each is different for different objects and this distinction is learned through demonstration data.
- "Go to the trash can", "go to the table", etc.
- "Place the coke", "place the 7up", etc.

Recall that we wish to find the affordance function $p(c_\pi|s,\ell_\pi)$, which indicates the probability of $c$-ompleting the skill with description $\ell_\pi$ successfully from state $s$. Our learned policies produce a Q-function, $Q^\pi(s,a)$. Given $Q^\pi(s,a)$ with action $a$ and state $s$, value $v(s) = \max_a Q^\pi(s,a)$ is found through optimization via the cross entropy method, similar to MT-Opt. For brevity below we refer to the value functions by their skill-text description $\ell_\pi$ as $v^{\ell_\pi}$ and the affordance function as $p_{\ell_\pi}^{\text{affordance}}$. Furthermore, SayCan enforces logic that if a skill that has already been completed and the reward received (e.g., navigating to the table the robot is already in front of) then it should not be performed. Due to artifacts of training and each implementation, the value functions require calibration to be directly applied as a probability. The parameters used for calibration are determined empirically. In practice we found this calibration fairly robust and straightforward and that the combination of the LLM and affordance function complement each other to reduce errors. For picking for instance, we see when an object is not present or when the robot is navigating, we find a consistent minimum value, while if the object is present the value rises quickly when the object is able to be picked, and peak consistently when the object is picked. For navigation, we set this value so that it is calibrated to the size of the scene.

- **Pick.** We find the trained value functions generally have a minimum value for when a skill is not possible and a maximum when the skill is successful and thus we normalize the value function to get a affordance function with

$$p_{\text{pick}}^{\text{affordance}} = \text{clamp}(\frac{v^{\text{pick}} - v_{\text{min}}^{\text{pick}}}{v_{\text{max}}^{\text{pick}} - v_{\text{min}}^{\text{pick}}}, 0, 1), \text{where } v_{\text{max}}^{\text{pick}} = 0.5, v_{\text{min}}^{\text{pick}} = 0.2.$$

- **Go to.** The affordance function of go to skills are based on the distance $d$ (in meters) to the location. We use

$$p_{\text{goto}}^{\text{affordance}} = \text{clamp}(\frac{d_{\text{max}}^{\text{goto}} - d^{\text{goto}}}{d_{\text{max}}^{\text{goto}} - d_{\text{min}}^{\text{goto}}}, 0, 1), \text{where } d_{\text{max}}^{\text{goto}} = 100, d_{\text{min}}^{\text{goto}} = 0.$$

- **Place.** We assume place is always possible, $p_{\text{place}}^{\text{affordance}} = 1.0$, since we find language is sufficient to understand place is only possible after a pick. In the future work having an affordance function module for place could further improve the performance of SayCan.

- **Terminate.** We give terminate a small affordance value, to make sure the planning process terminates when there is no feasible skills to choose from. $p_{\text{terminate}}^{\text{affordance}} = 0.1$.

### D.3   LLM Size

SayCan is able to improve with improved language models. The LLM used herein was PaLM [11], a 540B parameter model. In this section we ablate over 8B, 62B, and 540B parameter models as well as the 137B parameter FLAN model [86] which is finetuned on a "instruction answering" dataset. Table 3 shows each model on the generative problem. Table 4 shows PaLM 540B and FLAN on robot. We find that generally larger models perform better, though the difference between the 62B and 540B model is small. Results in other works, such as Chain of Thought Prompting [20], indicate this difference may be more pronounced on more challenging problems. We also find that PaLM outperforms FLAN. Though FLAN was finetuned on instruction answering, the broader and improved dataset for PaLM may make up for this difference in training.

### D.4   LLM Prompt

The LLM uses prompt engineering and a strict response structure to score skills. But, as SayCan as a whole requires affordances from a world embodiment, it is not straightforward to optimize this structure and tune parameters quickly. Thus we built a language-based simulator which, given a query and a solution sequence of skills, outputs affordances consistent with the query and solution. It also generates consistent distractor affordances to ensure robustness. The simulator then verifies that SayCan recovers the correct solution and tests how confident SayCan is in the correct solutions.

| Family | Num | PaLM 540B [11] | PaLM 62B | PaLM 8B | FLAN 137B [86] |
|---|---|---|---|---|---|
| NL Single | 15 | 87% | 73% | 20% | 40% |
| NL Nouns | 15 | 53% | 47% | 20% | 40% |
| NL Verbs | 15 | 93% | 100% | 60% | 87% |
| Structured | 15 | 100% | 100% | 67% | 73% |
| Embodiment | 11 | 36% | 27% | 27% | 0% |
| Crowd Sourced | 15 | 80% | 73% | 47% | 47% |
| Long-Horizon | 15 | 60% | 73% | 20% | 0% |
| Total | 101 | 74% | 72% | 38% | 43% |

Table 3: Ablations over the size of the LLM. Compared only with the generative outputs (no value function) with USE embeddings [8].

| Family | Num | PaLM [11] | | FLAN [86] | |
|---|---|---|---|---|---|
| | | Plan | Execute | Plan | Execute |
| NL Single | 15 | 100% | 100% | 67% | 67% |
| NL Nouns | 15 | 67% | 47% | 60% | 53% |
| NL Verbs | 15 | 100% | 93% | 80% | 67% |
| Structured | 15 | 93% | 87% | 100% | 87% |
| Embodiment | 11 | 64% | 55% | 64% | 55% |
| Crowd Sourced | 15 | 87% | 87% | 73% | 67% |
| Long-Horizon | 15 | 73% | 47% | 47% | 33% |
| Total | 101 | 84% | 74% | 70% | 61% |

Table 4: Success rates of instructions by family. SayCan achieves a planning success rate of 84% and execution success rate of 74% with PaLM and FLAN achieves 70% planning and 61% success. SayCan scales and improves with improved LLMs.

In Table 5 we test the effect of the number of examples in the prompt on the planning success rate in the language-based simulator (over 50 demonstrative instructions). We show a success rate with and without requiring the plan to terminate; without examples we found the LLM was unlikely to issue a "done" phase. With no examples SayCan is able to successfully plan 54% without the done condition, but only 10% with the done condition. Though it makes mistakes, clearly some information is already imbued within the language model. It is able to correctly solve "Can I have a redbull please?" and "Move the chips bag from the table to the counter.". With only one example the LLM quickly improves in both planning rates, though still fails to terminate the plan occasionally. After only four examples the LLM is performant, planning 82% of the queries correctly, though the remaining errors are largely within a single instruction family: Long-Horizon. Finally, the prompt used in this work, Listing 1, involved 17 examples and recovered 88% of the solutions correctly.

We note here briefly a few lessons learned in prompt engineering and structuring the final prompt. Providing explicit numbers between steps (e.g., 1., 2., instead of combining skills with "and then" or other phrases) improved performance, as did breaking each step into a separate line (e.g. adding a "\n" between steps). Examples which overly include objects used in the actual planning tend to bias results to those objects (e.g., if every example is about apples then the apple scoring will be off in planning). Phrasing of the natural language names of skills and objects is important due to the auto-regressive nature of the LLM scoring – skills and objects should be naturally named and errors such as misspellings or mismatches in "a" vs "an" can be problematic. Notably, since user generated instructions are taken as given such fragility is not issues for the input, allowing a robustness to user queries. For our language model, PaLM [11], structuring the interaction as dialog was both more natural and performant. Although dialog is used as prompt, the model generalized to imperative sentences at deployment time.

| Num Examples | Require Termination | No Termination Required |
|---|---|---|
| 0 | 10% | 52% |
| 1 | 64% | 74% |
| 2 | 68% | 76% |
| 4 | 82% | 84% |
| 8 | 80% | 80% |
| Full Prompt (17) | 88% | 88% |

Table 5: Number of prompt engineering examples versus planning rate (when requiring the plan to know when to terminate, e.g., issue a "done" command, or not).

Listing 1: Prompt in SayCan.

```
Robot: Hi there, I'm a robot operating in an office kitchen.
Robot: You can ask me to do various tasks and I'll tell you the sequence of actions I would do to
    accomplish your task.
Human: How would you hold the snickers?
Robot: 1. pick up the snickers, 2. done.
Human: How would you put the trailmix here?
Robot: 1. put down the trailmix, 2. done.
Human: How would you put a water bottle and an oatmeal next to the microwave?
Robot: 1. find a water bottle, 2. pick up the water bottle, 3. go to microwave, 4. put down the water
    bottle, 5. find an oatmeal, 6. pick up the oatmeal, 7. go to the microwave, 8. put down the oatmeal,
    9. done.
Human: How would you put a grapefruit from the table into the bowl?
Robot: 1. find a grapefruit, 2. pick up the grapefruit, 3. go to the bowl, 4. put down the grapefruit, 5.
    done.
Human: How would you get a sponge from the counter and put it in the sink?
Robot: 1. find a sponge, 2. pick up the sponge, 3. go to the sink, 4. put down the sponge, 5. done.
Human: How would you move the water bottle from the table to the counter?
Robot: 1. find a water bottle, 2. pick up the water bottle, 3. go to the counter, 4. put down the water
    bottle, 5. done.
Human: How would you bring me some snacks?
Robot: 1. find a twix, 2. pick up the twix, 3. bring it to you, 4. put down the twix, 5. find an dried
    fruit, 6. pick up the dried fruit, 7. bring it to you, 8. put down the dried fruit, 9. done.
Human: How would you bring me an orange?
Robot: 1. find an orange, 2. pick up the orange, 3. bring it to you, 4. put down the orange, 5. done.
Human: How would you deliver a can of coke to me?
Robot: 1. find a coke can, 2. pick up the coke can, 3. bring it to you, 4. put down the coke can, 5. done.
Human: I finished my drink, can you throw it away and bring me a mountain dew?
Robot: 1. find a sprite can, 2. pick up the sprite can, 3. go to the trash can, 4. put down the sprite can
    , 5. find a mountain dew can, 6. pick up the mountain dew can, 7. bring it to you, 8. put down the
    mountain dew can, 9. done.
Human: How would you let go of the banana?
Robot: 1. put down the banana, 2. done.
Human: How would you put the grapes in the bowl and then move the cheese to the table?
Robot: 1. find the grapes, 2. pick up the grapes, 3. go to the bowl, 4. put down the grapes, 5. pick up
    the cheese, 6. go to the table, 7. put down the cheese, 8. tell you I'm done.
Human: How would you pick up the peanuts?
Robot: 1. pick up the peanuts, 2. done.
Human: How would you bring me the peanuts?
Robot: 1. pick up the peanuts, 2. bring it to you, 3. put down the peanuts, 4. done.
Human: How would you throw away a coffee cup?
Robot: 1. find a coffee cup, 2. pick up the coffee cup, 3. go to trash can, 4. put down the coffee cup, 5.
    done.
Human: How would you place a knife and a banana to the table?
Robot: 1. find a knife, 2. pick up the knife, 3. go to the table, 4. put down the knife, 5. find a banana,
     6. pick up the banana, 7. go to the table, 8. put down the banana, 9. done.
```

# E  Experiments

## E.1  Tasks

Below we include every instruction run, which environment it was run in, and its planning and execution success rate. Table 5 shows all instructions as broken down by instruction family, listed below and initially defined in Section 4 Table 1.

- **Natural Language (NL) Single Primitive.** Given a natural language command corresponding to performing a single primitive, can SayCan recover that primitive skill and terminate?
- **NL Noun.** Given a natural language query that replaces a noun (typically an object or location) with a synonym, can SayCan execute the appropriate sequence?

- **NL Verbs.** Given a natural language query that replaces a verb (typically an action) with a synonym, can SayCan execute an appropriate sequence?

- **Structured Language.** Given a structure language query that mirrors the NL Verbs and spells out the sequence of commands, how well can SayCan plan compared to NL Verbs? This acts as an ablation to see the performance loss of understanding a natural language query over an explicit solution.

- **Embodiment.** Given a query with different environment and robot states, can SayCan still execute at a high rate? This tests the performance of SayCan's affordance model and the LLM's ability to reason within it.

- **Crowd-Sourced.** These queries were crowd sourced from Mechanical Turk by giving humans a description of what occurred (e.g., an apple was moved in front of you) and asking them what they would ask the robot to do. They were also crowd sourced by asking humans in a real office kitchen to command the robot to perform tasks (given knowledge of the robot's abilities). This tests SayCan's performance with natural requests.

- **Long-Horizon.** These challenging queries require SayCan to reason over temporally extended instructions to investigate how well it scales to such regimes.

### E.2 Adding Skills: Drawer Manipulation

In order to support drawer manipulation we added another category of skills in SayCan.

- **Drawer Manipulation.** For drawer manipulation we use the learned policies in Appendix C and Section 3 with actions from BC and value functions from heuristics (If the robot is next to the drawer, all drawer tasks are possible). In natural language these are specified as "open the drawer", "close the drawer" and "put the object in the drawer".

A few drawer-specific prompts also need to be added to teach the robot how to chain the drawer skills together. The prompts are shown in Listing 2.

Listing 2: Drawer Prompt in SayCan.

```
Human: open the drawer
Robot: 1. go to the drawers, 2. open the drawer, 3. done.
Human: restock orange juice into the drawer
Robot: 1. go to the drawers, 2. open the drawer, 3. put orange juice in the drawer, 4. close the drawer,
    5. done.
Human: restock two bottles of orange juice into the drawer
Robot: 1. go to the drawers, 2. open the drawer, 3. put orange juice in the drawer, 4. put orange juice in
    the drawer, 5. close the drawer, 6. done.
```

The results of the drawer tasks are shown in Table. 6. SayCan achieved an overall planning success rate of 100% and execution success rate of 33%. The main failure cases are manipulation failures,

| Instruction |
| --- |
| How would you pick up the coke can |
| How would you put the coke can in the your gripper |
| How would you grasp the coke can |
| How would you hold onto the coke can |
| How would you lift and hold the coke can up |
| How would you put the coke can down |
| How would you place the coke can on the table |
| How would you let go of the coke can |
| How would you release the coke can |
| How would you place the coke can |
| How would you move to the table |
| How would you go to the table |
| How would you park at the table |
| How would you come to the table |
| How would you navigate to the table |

(a) NL Single Primitive

| Instruction |
|---|
| How would you throw away the apple |
| How would you bring me a sponge? |
| How would you bring me a coke can |
| How would you grab me an apple |
| How would you grab me a 7up from the table |
| How would you deliver the red bull to the close counter |
| How would you throw away the jalapeno chips |
| How would you restock the rice chips on the far counter |
| How would you recycle the coke can |
| How would you throw away the water bottle |
| How would you bring me something hydrating |
| How would you put the apple back on the far counter |
| How would you recycle the 7up |
| How would you throw away jalapeno chips |
| How would you compost the apple |

(b) NL Verb

| Instruction |
|---|
| How would you bring me lime drink |
| How would you bring me something to clean the kitchen with |
| How would you bring me something to eat |
| How would you put the grapefruit drink on the close counter |
| How would you move the sugary drink to the far counter |
| How would you move something with caffine from the table to the close counter |
| How would you bring me an energy bar |
| How would you bring me something to quench my thirst |
| How would you bring me a fruit |
| How would you bring me a fruit from the close counter |
| How would you bring me something that is not a fruit from the close counter |
| How would you bring me a soda from the table |
| How would you bring me a soda |
| How would you bring me a bag of chips from close counter |
| How would you bring me a snack |

(c) NL Nouns

| Instruction |
|---|
| How would you pick up the apple and move it to the trash |
| How would you pick up the sponge and bring it to me |
| How would you pick up the coke can and bring it to me |
| How would you pick up the apple and bring it to me |
| How would you pick up the 7up and bring it to me |
| How would you pick up the redbull and move it to the close counter |
| How would you pick up the jalapeno chips and move it to the trash |
| How would you pick up the rice chips and move it to the far counter |
| How would you pick up the coke can and move it to the trash |
| How would you pick up the water bottle and move it to the trash |
| How would you pick up the grapefruit soda and bring it to me |
| How would you pick up the apple and move it to the far counter |
| How would you pick up the 7up and move it to the trash |
| How would you pick up the jalepeno chips and move it to the trash |
| How would you pick up the apple and move it to the trash |

(d) Structured Language

| Instruction |
|---|
| How would you put the coke can down on the far counter(with operator) |
| How would you put the coke can down on the far counter(at table) |
| How would you put the coke can down on the far counter(at table with coke can in hand) |
| How would you put the coke can down on the far counter(at far counter with coke can in hand) |
| How would you put the sponge on the close counter(with operator) |
| How would you put the sponge on the close counter(at far counter) |
| How would you put the sponge on the close counter(at far counter with sponge in hand) |
| How would you put the sponge on the close counter(at close counter with coke can in hand) |
| How would you pick up the drink from the far counter |
| I left something on the table, can you throw it away? |
| I left something on the table or the counter, can you bring it to me? |

(e) Embodiment

| Instruction |
|---|
| I opened a pepsi earlier. How would you bring me an open can? |
| I spilled my coke, can you bring me a replacement? |
| I spilled my coke, can you bring me something to clean it up? |
| I accidentally dropped that jalapeno chip bag after eating it. Would you mind throwing it away? |
| I like fruits, can you bring me something I'd like? |
| There is a close counter, far counter, and table. How would you visit all the locations? |
| There is a close counter, trash can, and table. How would you visit all the locations? |
| Redbull is my favorite drink, can I have one please? |
| Would you bring me a coke can? |
| Please, move the pepsi to the close counter |
| Please, move the ppsi(intentional typo) to the close cuonter |
| Can you move the coke can to the far counter? |
| Can you move coke can to far counter? |
| Would you throw away the bag of chips for me? |
| Would you throw away the bag of chpis(intentional typo) for me? |

(f) Crowd-Sourced

| Instruction |
|---|
| How would you put an energy bar and water bottle on the table |
| How would you bring me a lime soda and a bag of chips |
| Can you throw away the apple and bring me a coke |
| How would you bring me a 7up can and a tea? |
| How would throw away all the items on the table? |
| How would you move an multigrain chips to the table and an apple to the far counter? |
| How would you move the lime soda, the sponge, and the water bottle to the table? |
| How would you bring me two sodas? |
| How would you move three cokes to the trash can? |
| How would you throw away two cokes? |
| How would you bring me two different sodas? |
| How would you bring me an apple, a coke, and water bottle? |
| I spilled my coke on the table, how would you throw it away and then bring me something to help clean? |
| I just worked out, can you bring me a drink and a snack to recover? |
| How would you bring me a fruit, a soda, and a bag of chips for lunch |

(g) Long-Horizon

Table 5: **List of all instructions** We evaluate the algorithm on 101 instructions on 2 scenes. The metrics and success definitions can be found in Sec. 4.

where the robot fails to open the drawer wide enough to put objects in it, or fails to completely close the drawer.

| Instruction | Plan rate | Execution rate |
|---|---|---|
| restock the coke and pepsi into the drawer | 1.0 | 0.0 |
| hide the 7up in the drawer | 1.0 | 0.83 |
| restock the coke into the drawer | 1.0 | 0.17 |

Table 6: Drawer tasks

### E.3 Chain of Thought Reasoning

One limitation of vanilla SayCan is that it doesn't handle tasks that involves negation. This is inherited from underline language models, and studied in the NLP community [14]. However, we found by using chain-of-thought prompting [20] we can improve SayCan on this front.

For chain-of-thought prompting-based SayCan, we need to modify the prompt to include a part called "Explanation". We also slightly change how we use the language model. Instead of directly using the scoring interface to rank possible options, we first use the generative decoding of LLM to create an explanation, and then use the scoring mode, by including the explanation into the prompt. The full prompt is shown in Listing 3.

Listing 3: Chain-of-thought prompt in SayCan.

```
Robot: Hi there, I'm a robot operating in an office kitchen.
You can ask me to do various tasks and I'll tell you the sequence of actions I would do to accomplish your
    task.
The following objects are in the scene: 7up, apple, tea, multigrain chips, kettle chips, jalapeno chips,
    rice chips, coke, grapefruit soda, pepsi, redbull, energy bar, lime soda, sponge, and water bottle.
The following locations are in the scene: close counter, far counter, table, you, trash, bowl.
Human: Hold the sponge
Explanation: The user has asked for a single skill, I will just pick up the sponge.
Robot: 1. pick up the sponge, 2. done.
Human: Put the kettle chips here
Explanation: The user has asked for a type of chips, I will bring those to them.
Robot: 1. put down the kettle chips, 2. done.
Human: Move the grapefruit drink from the table to the close counter
Explanation: The user has asked me to move the grapefruit drink to the counter.
Robot: 1. find a grapefruit soda, 2. pick up the grapefruit soda, 3. go to the counter, 4. put down the
    grapefruit soda, 5. done.
Human: Bring me some snacks
Explanation: The user has asked for snacks, I will bring jalapeno chips and an apple.
Robot: 1. find a jalapeno chips, 2. pick up the jalapeno chips, 3. bring it to you, 4. put down the
    jalapeno chips, 5. find an apple, 6. pick up the apple, 7. bring it to you, 8. put down the apple, 9.
     done.
Human: Bring me something that isn't a fruit
Explanation: The user has asked for something food that isn't an fruit, I will bring an energy bar.
Robot: 1. find an energy bar, 2. pick up the energy bar, 3. bring it to you, 4. put down the energy bar,
    5. done.
Human: Put the rice chips in the bowl and then move the tea to the table
Explanation: The user has asked me to do two tasks, I will do one and then the other.
Robot: 1. find the rice chips, 2. pick up the rice chips, 3. go to the bowl, 4. put down the rice chips,
    5. pick up the tea, 6. go to the table, 7. put down the tea, 8. done.
Human: Throw away a redbull
Explanation: The user has asked me to throw away the redbull can, I will move it to the trash.
Robot: 1. find a redbull, 2. pick up the redbull, 3. go to trash, 4. put down the redbull, 5. done.
```

A few successful rollouts of the model at evaluation time is shown in Listing 4. As we can see, with chain of thought prompting, the model can handle negations and tasks that require reasoning.

Listing 4: Chain-of-thought planning rollout with SayCan.

```
Human: Can you bring a drink without caffeine but has some fruit flavor?
Explanation: The user has asked for a drink without caffeine but has some fruit flavor, I will bring the
    lime soda.
Robot: 1. find a lime soda, 2. pick up the lime soda, 3. bring it to you, 4. done

Human: Bring me chips that taste spicy
Explanation: The user has asked for chips that taste spicy, I will bring the Jalapeno chips.
Robot: 1. find the jalapeno chips, 2. pick up the jalapeno chips, 3. bring it to you, 4. put down the
    jalapeno chips, 5.done

Human: I do not like pepsi, bring me something similar
Explanation: The user has asked for something similar to coke, I will bring coke
Robot: 1. find a coke, 2. pick up the coke, 3. bring it to you, 4. put down the coke, 5. done

Human: The snack is too small, bring me something more filling
Explanation: The user has asked for a snack that is more filling, I will bring the multigrain chips
Robot: 1. find the multigrain chips, 2. pick up the multigrain chips, 3. bring it to you, 4. put down the
    multigrain chips, 5. done
```

### E.4 Multilingual Queries

Since the underlying LM we used [11] has been trained on multilingual corpora, SayCan can handle multilingual queries out of the box. The results of SayCan on multilingual queries are summarized in Table. 7, and there is almost no performance drop on planning success rate when chaning the queries from English to Chinese, French and Spanish.

| Instruction | Plan rate |
|---|---|
| bring me a can of coke | 1.0 |
| throw away the coke can | 1.0 |
| I spilled my coke, can you bring me something to help clean | 1.0 |
| 拿一罐可乐给我 | 1.0 |
| 扔掉可乐罐 | 1.0 |
| 我的可乐洒了，你能给我拿点东西来帮忙打扫吗 | 1.0 |
| apporte moi une canette de coca | 1.0 |
| jeter la canette de coca | 1.0 |
| J'ai renversé mon coca, peux-tu m'apporter quelque chose pour m'aider à nettoyer | 0.0 |
| tráeme una lata de coca cola | 1.0 |
| tirar la lata de coca cola | 1.0 |
| Derramé mi coca cola, ¿puedes traerme algo para ayudar a limpiar | 1.0 |

Table 7: Multilingual queries plan success rate. instruction 4-12 are the Chinese, French and Spanish translation of first 3 queries.

### E.5 Additional Results

Additional results are shown in Figure 12 and Figure 13 and some failure cases in Figure 14. For videos of the rollouts, please visit the anonymous website `https://saycan-corl.github.io`

## F Open Source Environment

We have open sourced an implementation of SayCan in a Google Colab notebook for a tabletop robot. Code can be found here. The environment is shown in Figure 16 and is a tabletop with a UR5 robot and randomly generated sets of colored blocks and bowls. It is implemented with a ViLD object detector for affordances [90] and GPT-3 as the large language model [2]. Steps are output in the form `robot.pick_and_place(object, location)`, leveraging the ability of LLMs to output code structures. The policy is implemented with CLIPort [48], which is trained to output a pick and place location.

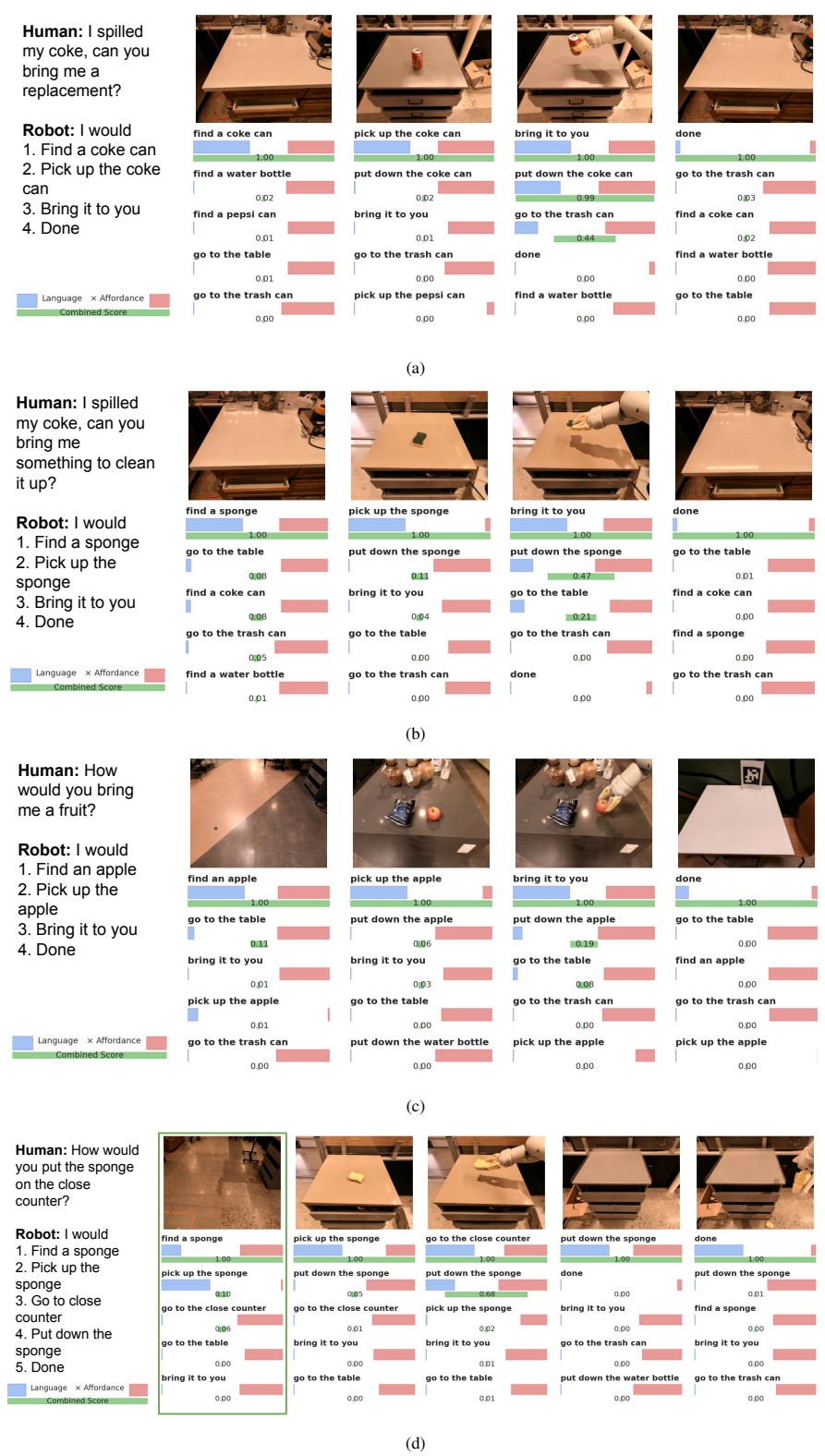

Figure 12: Visualization of the decision making process of SayCan shows its interpretability and successful temporally extended execution, where the top combined score chooses the correct skill.

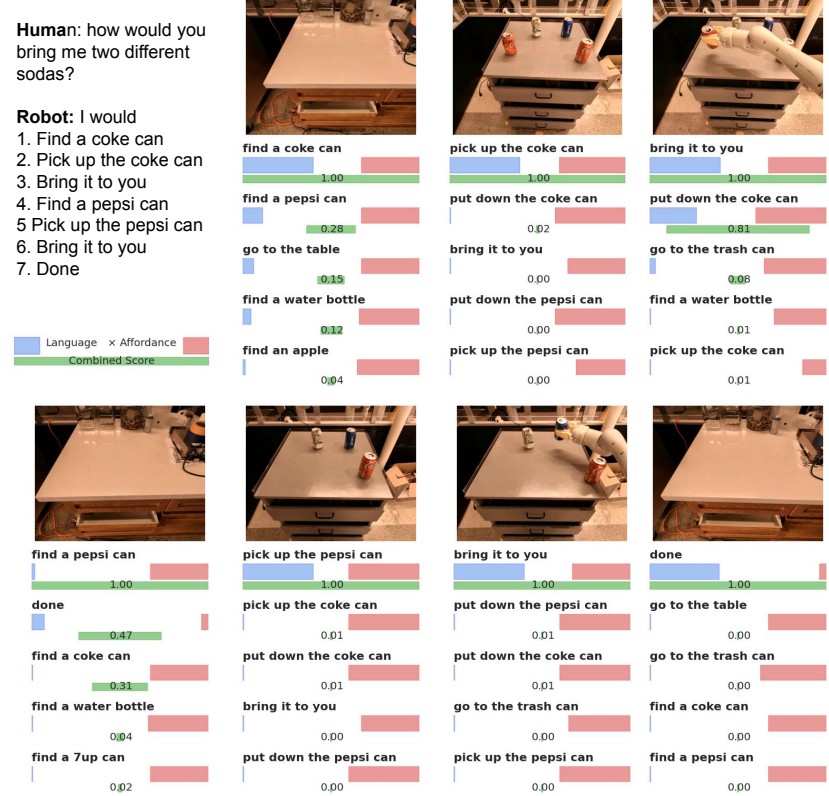

(a) In this long-horizon task, the language model gives high score to the two sodas. After the coke is delivered, the language model scores pepsi higher. The affordance rating overcomes potential early termination after the first can has been delivered.

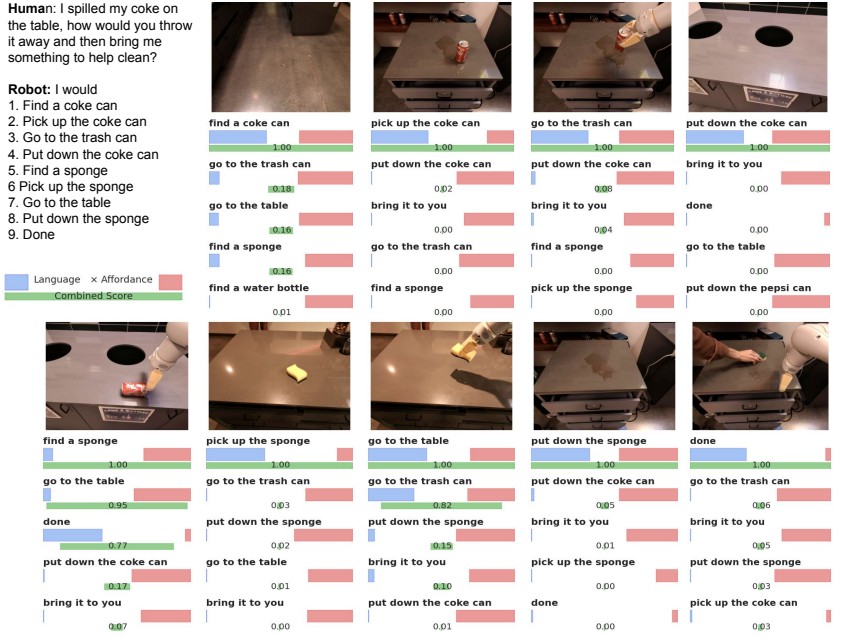

(b) In this task, the model completes a 9-step plan. It narrowly avoids an early termination at step 5.

Figure 13: Long horizon sequences, see the video on our website `say-can.github.io` for more.

**Huma**n: How would you bring me a bag of chips from close counter?

**Robot:** I would
1. Go to the close counter
2. Go to the far counter
3. Go to the close counter
4. Go to the far counter …

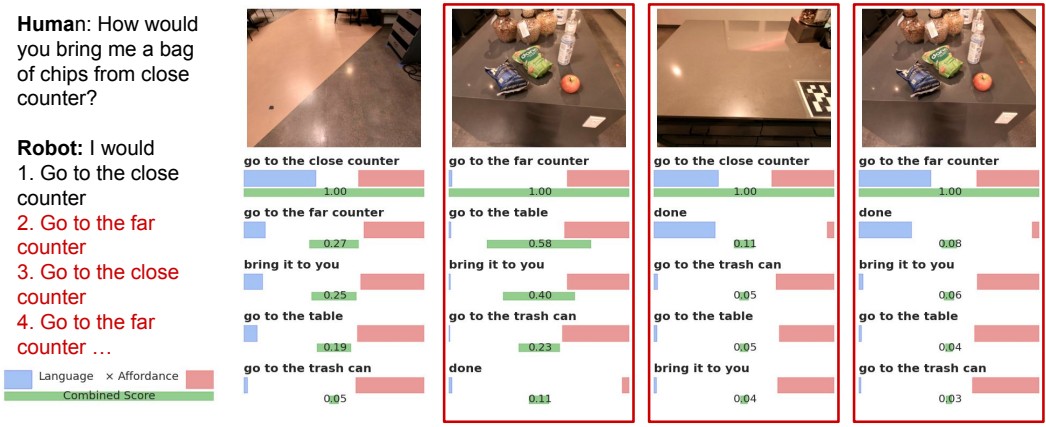

(a) The affordance model fails to identify either bag of chips as pickable, though the language model approaches the counter twice.

**Human:** How would you bring me something to clean the kitchen with?

**Robot:** I would
1. Find a sponge
2. Put down the sponge
3. Bring it to you
4. Done

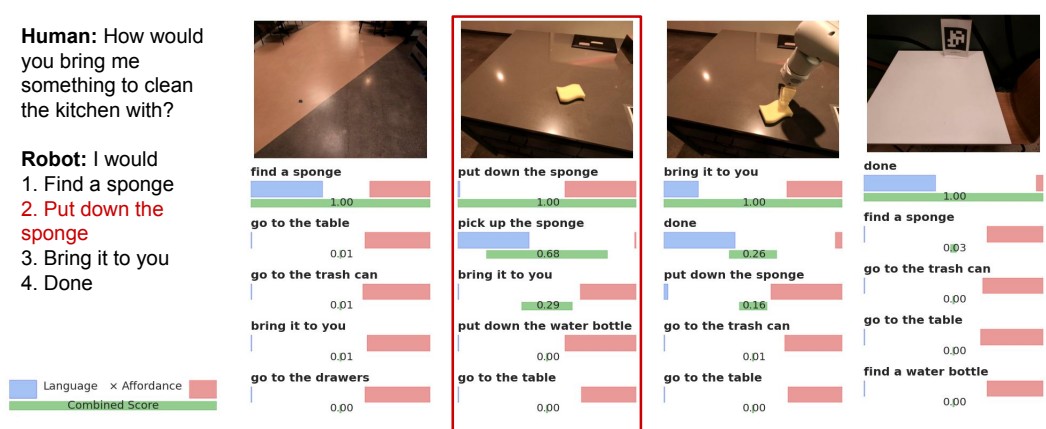

(b) The affordance model fails to identify the sponge as pickable.

**Huma**n: How would you bring me a lime soda and a bag of chips?

**Robot:** I would
1. Find a lime soda
2. Pick up the lime soda
3. Bring it to you
4. Done

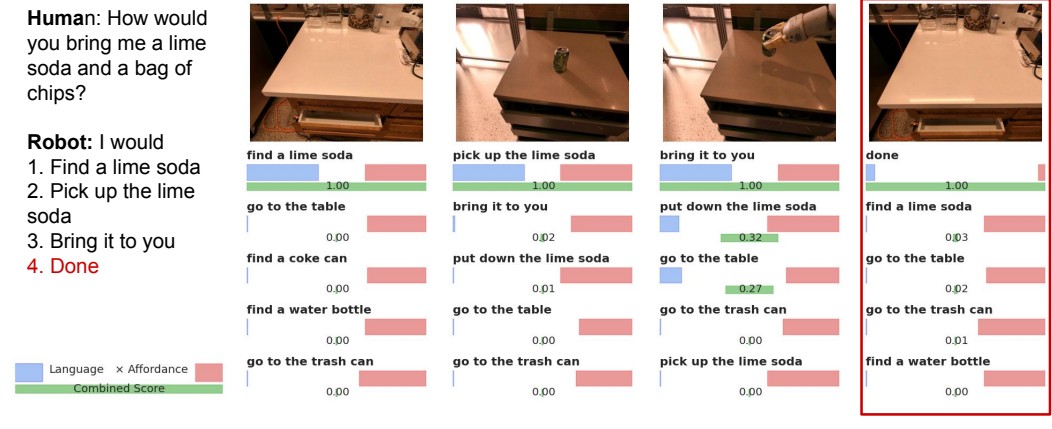

(c) Language model terminates a long-horizon task prematurely.

Figure 14: Failure cases. The planning success rate was 84%. Of the errors, 65% were a result of an LLM error and 35% were affordance errors.

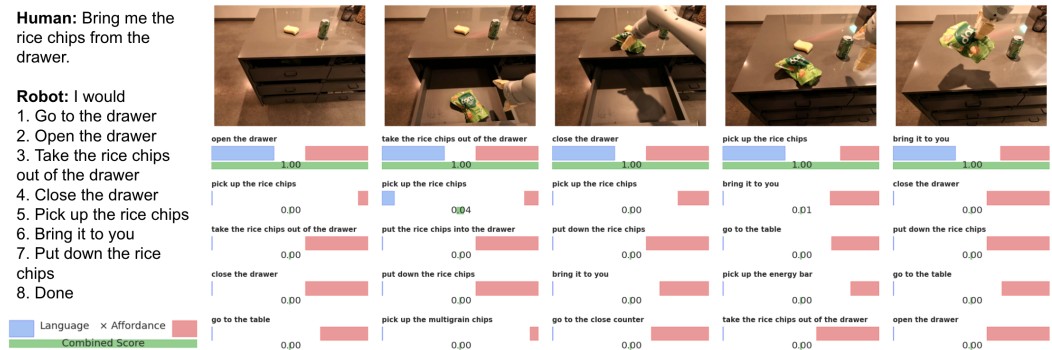

Figure 15: A sequence of the robot taking rice chips from the drawers. Note the robot only has one arm, so it needs to plan a long sequence to first take rice chips out of drawer and place on the counter, and then pick it up again after closing the drawer.

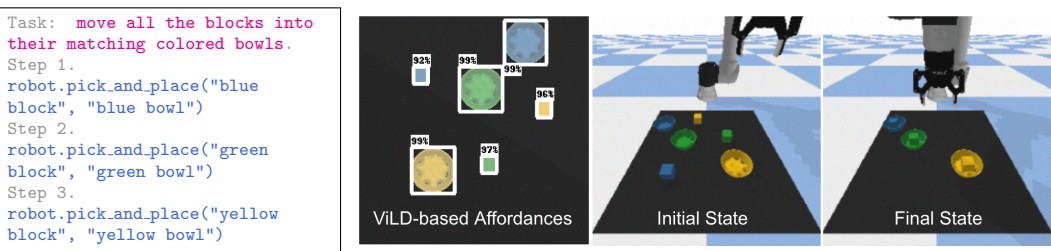

Figure 16: We have open sourced a Colab with a tabletop environment, a UR5 robot, and CLIPort-based policy here: https://github.com/google-research/google-research/blob/master/saycan/SayCan-Robot-Pick-Place.ipynb.

