# OpenReview forum: "Do As I Can, Not As I Say: Grounding Language in Robotic Affordances"
_robot-learning.org/CoRL/2022/Conference — CoRL 2022 Oral_

### Official Review · Reviewer_W6E5 · 2022-07-28

**Originality:** Very Good
**Technical Quality:** Very Good
**Clarity Of Presentation:** Excellent
**Impact:** 4

**Recommendation:**

Strong Accept: I recommend accepting the paper and will argue for my recommendation even if other reviewers hold a different opinion.

**Summary:**

This paper proposes a method for robots to understand the instructions expressed in natural language and execute them in the real world. The sequences of low-level actions to perform instructions are estimated by using a large language model (LLM), and an appropriate prompt that extracts low-level action representations of instructions from the LLM is proposed. Furthermore, a learned value function is used to estimate how likely each action is to succeed in the current state in the real world. Combining probability provided by LLM and value functions, appropriate actions in the current state can be selected in actions that are predicted as next step actions by LLM. By appending the action representation to the prompt and querying the LLM, further next action can be predicted. Experimental results show that the proposed method can work well in the real environment.

**Issues:**

Described in the strengths and weaknesses section.

**Quality Of The Limitations Section:**

Limitations are addressed clearly

**Reviewer Expertise:**

3: The reviewer is fairly confident that the evaluation is correct

**Robotics Focus:**

Sufficient demonstration on hardware

**Strengths And Weaknesses:**

# Strengths
- A novel method for robots to understand the instructions using LLM is proposed.
- Experiment was conducted in the real world and the results show that the proposed method can work well in the real environment.

# Weaknesses and suggestions
- A lot of important information is written in the appendix. I understand due to space limitations but the contents should be selected and the important results should be written in body text.
- Line 179: What is the definition of `skills` and `skill families`


**Summary Of Recommendation:**

I think it is worth accepting this paper because it presents a new potential of the use of the LLMs in robotics.

---

> ### Author Response · Authors · 2022-08-23
> **Thank you for your review**
>
> Thank you for your review and feedback. We have carefully reviewed the paper to ensure the most relevant information is in the main body of the paper. We have also appended the Appendix to the end of the main paper PDF for clarity and the ability to link sections. Lastly, we have open sourced a version of SayCan with a simulated tabletop robot ([Colab](https://colab.sandbox.google.com/github/saycan-corl/saycan-corl.github.io/blob/main/SayCan-Robot-Pick-Place.ipynb)) to help further clarify important details.
>
> > Line 179: What is the definition of skills and skill families
>
> We have clarified L179: “Inspired by common skills one might pose to a robot in a kitchen environment, we propose 551 skills that span seven skill families and 17 objects. The skill families are picking, placing, and rearranging objects, placing objects in specific configurations, opening and closing drawers, and navigating to various locations. A skill may then be “pick up the apple” or “go to the trash can”.”

---

> > ### Comment · Reviewer_W6E5 · 2022-08-26
> > **Thank you for your response**
> >
> > Thank you for your response. The Colab implementation is beneficial for readers. I think that this paper is "Strong Accept."

---

### Official Review · Reviewer_h3Ar · 2022-08-01

**Originality:** Excellent
**Technical Quality:** Excellent
**Clarity Of Presentation:** Excellent
**Impact:** 4

**Recommendation:**

Strong Accept: I recommend accepting the paper and will argue for my recommendation even if other reviewers hold a different opinion.

**Summary:**

By assuming a set of primitives and their learned affordances, SayCan reweighs an LLM's generated steps in a task plan by their affordances. This enables a specific robot hardware to influence the LLM planner by only appended chosen steps after weighing by affordances.

**Issues:**

- This work assumes the robot is given a set of skills. Appendix C.2 that choosing this set of primitive requires 1) a set of candidate tasks to see which primitives would be useful for the LLM, and 2) an attempt in training their affordance-based policy to gauge their real world consistency and success. I think this is a limitation which could be discussed in more detail in the main paper.

**Quality Of The Limitations Section:**

Limitations are addressed clearly

**Reviewer Expertise:**

5: The reviewer is absolutely certain that the evaluation is correct and very familiar with the relevant literature

**Robotics Focus:**

Sufficient demonstration on hardware

**Strengths And Weaknesses:**

 - Excellent figures, real world experiment results, and video.
 - Simple and effective key idea.

**Summary Of Recommendation:**

This paper demonstrates an impressive zero-shot, language-conditioned mobile manipulation system, capable of performing long horizon tasks in a robot hardware aware manner. The discussion of failure modes and challenges in building the system in the main paper and appendix were useful. The quantitative and qualitative evaluations were comprehensive. The key idea is simple and effective, and leads to a useful and plan-interpretable system.

---

> ### Author Response · Authors · 2022-08-23
> **Thank you for your review**
>
> Thank you for your review and feedback. We’ve added additional discussion of this limitation in the conclusions: “... the primary bottleneck of the system is in the range and capabilities of the underlying skills. Beyond these skills, SayCan relies on reliable affordances to gauge the ability of the robot to perform a task given the state. Though with RL trained policies, the value function provides a reasonable approximation of the affordance, it is an open question how to best train affordances for general skills.”

---

> ### Comment · Reviewer_h3Ar · 2022-08-26
> **Final Decision**
>
> Thank you for the addition to the paper. This work is an impressive system. My final decision remains a `Strong Accept`.

---

### Official Review · Reviewer_Hat5 · 2022-08-01

**Originality:** Very Good
**Technical Quality:** Good
**Clarity Of Presentation:** Good
**Impact:** 4

**Recommendation:**

Strong Accept: I recommend accepting the paper and will argue for my recommendation even if other reviewers hold a different opinion.

**Summary:**

This paper introduces a method to map high-level natural-language instructions to skills that a mobile manipulator can execute in its environment. Given an instruction, an LLM is used to predict a sequence of skills and the probability of each skill helping to complete the instruction. An affordance function is used to predict whether a given skill can be executed at the current environment state.

**Issues:**

Besides the comments above, I have a few minor suggestions.
1. I think the authors could provide examples of the prompts in the main text since these prompts have very specific structures that I imagine are very important.
2. Why is the execution success rate for language ablations and planning success rate for value function ablations not included? I am very curious to see the execution success rate for the “Generative” baseline.

**Quality Of The Limitations Section:**

Additional details required

**Reviewer Expertise:**

4: The reviewer is confident but not absolutely certain that the evaluation is correct

**Robotics Focus:**

Sufficient demonstration on hardware

**Strengths And Weaknesses:**

Strengths:
1. Similar to the concurrent work, this paper introduces a general framework that uses LLMs to map high-level natural language instruction to low-level actions. Since the LLMs are trained on a large amount of text data, they contain enough task and object knowledge to predict how to perform different high-level tasks.
2. In addition to using LLMs to predict low-level actions, the actions are selected based on whether they are feasible given the capabilities of the robot and the environment states. This formulation provides an easy and intuitive way to ground abstract plans generated by the LLMs.
3. The feedback to the LLM is provided by appending actions that have been successfully executed to the prompt.
4. The framework is evaluated in both simulations and in a real-world kitchen with a mobile manipulator. Human raters are used to assess whether the tasks are planned successfully and executed correctly by the robots. Using human raters can account for the fact that many tasks have different ways to be completed.

Weakness:
I appreciate the fact that this paper introduces a very general framework that can potentially leverage different LLMs and primitive skills. However, I am worried that the main text lacks enough clarity in stating the assumptions and limitations of the current implementation.

1. What are the 551 skills? It would be great if the authors can explain the definition of skills in the context of this paper. For example, is grasping an apple a different skill than grasping an orange?
2. The main text mentions that the value function of a skill is trained via TD backups regardless of how a skill is implemented. However, this description contradicts the appendix stating that some of the value functions are defined based on heuristics. Another important detail missing from the main text is that the value functions are calibrated empirically. I imagine that this calibration is very important as the probabilities/likelihoods from the LLMs and affordance functions are not normalized. The authors should provide more details on calibrating the likelihoods and balancing the likelihoods from the LLMs and affordance functions in the main text. I also see examples (e.g., third step in Figure 12 (a) in the appendix) where the LLM and affordance function predict the highest probabilities for different actions, resulting in a very small margin between the combined probability of the correct action and of the incorrect one. How likely can this happen and cause failures?
3. Although the goal of this paper is to leverage the generalization capability of the LLMs, the tasks tested seem to be very similar. Most tasks can be completed with the sequence of <find obj, pick up obj, go to loc, and put down obj>. Therefore, it is hard to see if the LLMs are actually required to generalize. Why is simply predicting the target object not enough? (since the location of each object is also predefined as stated in the appendix). Another question is why does the location of each object need to be pre-specified instead of being predicted by the LLMs?
4. The only connection between the environment and the LLM is the action feedback. Therefore the LLM does not have direct access to the environment state. It would be interesting to see a discussion on this limitation.
5. Related to the comment above, I also hope to see a comparison (even just in simulation) with existing vision-and-language navigation/object interaction methods that finetune LLMs on specific tasks. This comparison would provide insight into whether prompting or finetuning is a more effective way to balance generalization and task success.



**Summary Of Recommendation:**

This paper proposes a novel idea of combining LLMs and value functions for planning. The idea is tested with different LLMs and different implementations of primitive skills in relatively diverse settings and on real robots. This paper can be improved more by discussing the assumptions that the current implementation holds.

---

> ### Author Response · Authors · 2022-08-23
> **Thank you for your review, we have addressed your comments**
>
> Thank you for your review and detailed feedback, which we believe has helped improve the paper.
>
> > 1. What are the 551 skills
>
> Each skill is an atomic text command that can be performed by the robot’s low-level policies. The skills were of the skill families: pick up object, place object, move object near object, knock over object, open drawers, close drawers, and go to location or object. A skill operates over the objects and locations shown in Figure 3. Thus a single skill may be “pick up the apple” or “go to the trash can”. Appendix Section C.2 discusses in more details the skills used for planning, as not all learned skills were applicable for long-horizon planning tasks.
>
> We have clarified on L179: “Inspired by common skills one might pose to a robot in a kitchen environment, we propose 551 skills that span seven skill families and 17 objects. The skill families are picking, placing, and rearranging objects, placing objects in specific configurations, opening and closing drawers, and navigating to various locations. A skill may then be “pick up the apple” or “go to the trash can”.”
>
> And Appendix Section C.2, now states “Each skill is thus an explicit text command performed by a low-level policy for that skill. For object above, we use the objects shown in Figure 3 (e.g., coke can, water bottle, jalapeno chips, apple, sponge, etc.). These objects can be placed in random configurations at locations throughout the scene. For location above, we consider several named locations with known positions shown in Figure 3 (e.g., table, trash can, etc.). This results in skills such as the following: …” and then a list of a few explicit skills.
>
> > 2. The main text mentions that the value function of a skill is trained via TD backups regardless of how a skill is implemented. However, this description contradicts the appendix stating that some of the value functions are defined based on heuristics. Another important detail missing from the main text is that the value functions are calibrated empirically. I imagine that this calibration is very important as the probabilities/likelihoods from the LLMs and affordance functions are not normalized. The authors should provide more details on calibrating the likelihoods and balancing the likelihoods from the LLMs and affordance functions in the main text.
>
> Thank you for noting this, we have expanded our discussion. SayCan is capable of incorporating many different policies and affordance functions through its probability interface, so it is not limited to just learned value functions – we believe this is a promising feature of SayCan such that arbitrary skills with affordances can be included. For calibration, in practice we found this one-time procedure to be robust and straightforward. For picking, we see that the policies maintain consistent low (but non-zero) values when the object is not in front of the robot or when the robot is navigating. Once the object is present and able to be picked, this value quickly rises and stabilizes once the pick is successful. Furthermore, as the affordance value works in concert with the LLM score, the two can help rectify errors and result in a robust process. We have included the above discussion in Appendix C.2.
>
> > I also see examples (e.g., third step in Figure 12 (a) in the appendix) where the LLM and affordance function predict the highest probabilities for different actions, resulting in a very small margin between the combined probability of the correct action and of the incorrect one. How likely can this happen and cause failures?
>
> We do see occasional errors such as the near miss seen in Figure 12a. L248 mentions that of the 16% planning errors, about two-thirds are a result of language model errors.
>
> > 3. Although the goal of this paper is to leverage the generalization capability of the LLMs, the tasks tested seem to be very similar. Most tasks can be completed with the sequence of <find obj, pick up obj, go to loc, and put down obj>. Therefore, it is hard to see if the LLMs are actually required to generalize. Why is simply predicting the target object not enough? (since the location of each object is also predefined as stated in the appendix).
>
> This is a good point and to show that this is not a limitation, we show drawer skills in Section 4.2 that require plans that include sequences such as "open drawer”, “pick object out of drawer”, “place object on counter”, “close drawer”, “pick object” and the reverse to put objects into the drawer. We also include plans that do not require steps like “find” and “put down” to show how embodiment affects the plans. We hope that these demonstrate both the expressiveness and extensibility of SayCan, and particularly the LLM’s ability to reason over more complex tasks. We have expanded our discussion of such sequences in Section 4.2 and added Figure 6. In general we find that we are skill limited rather than planning limited by the LLM as discussed in L329.

---

> > ### Author Response · Authors · 2022-08-23
> > **continued**
> >
> > > Another question is why does the location of each object need to be pre-specified instead of being predicted by the LLMs?
> >
> > We believe this is an interesting question and LLM is likely to contain significant semantic knowledge over where potential objects can be found, but we feel this is a problem more aligned with algorithms for object navigation and low-level learned policies than for the planning proposed here. SayCan is a general framework that, if given an object navigation policy, can easily incorporate it into planning.
> >
> > > 4. The only connection between the environment and the LLM is the action feedback. Therefore the LLM does not have direct access to the environment state. It would be interesting to see a discussion on this limitation.
> >
> > This is a valid limitation and important point. We have added the following discussion to our conclusions. “Finally, a limitation of the proposed framework is that the LLM only receives feedback from the environment through the selected skills, which can be limited, e.g., if a skill fails. It would be useful to investigate to what degree sources of environment feedback can be incorporated, such as through success detectors, scene descriptions, direct visual feedback, or visual-language models.”
> >
> > > 5. Related to the comment above, I also hope to see a comparison (even just in simulation) with existing vision-and-language navigation/object interaction methods that finetune LLMs on specific tasks. This comparison would provide insight into whether prompting or finetuning is a more effective way to balance generalization and task success.
> >
> > We appreciate that this comparison will be insightful and we will add a comparison for planning success rate on the generative task shown in Table 3. This will be added to the final version of the paper.
> >
> > > The main text lacks enough clarity in stating the assumptions and limitations of the current implementation.
> >
> > We have reviewed and edited the main to include necessary information and added an open source version ([Colab](https://colab.sandbox.google.com/github/saycan-corl/saycan-corl.github.io/blob/main/SayCan-Robot-Pick-Place.ipynb)) to further clarify the approach.
> >
> > > I think the authors could provide examples of the prompts in the main text since these prompts have very specific structures that I imagine are very important.
> >
> > We have added a reference to the prompt on L196 to link to the prompt, however for space considerations we do not think it will be possible to include the full prompt other than mentioning they are “example plans”.
> >
> > > Why is the execution success rate for language ablations and planning success rate for value function ablations not included? I am very curious to see the execution success rate for the "Generative" baseline.
> >
> > We do not include execution rate for these because getting execution rate requires significant testing on robot (running 101 long-horizon queries in each environment) and we expect that the ratio between planning and execution (88% for SayCan) to be consistent or lower for these queries.

---

### Official Review · Reviewer_JdHL · 2022-08-04

**Originality:** Very Good
**Technical Quality:** Very Good
**Clarity Of Presentation:** Very Good
**Impact:** 3

**Recommendation:**

Weak Accept: I recommend accepting the paper, but will not argue for my recommendation if the majority of other reviewers have a different opinion.

**Summary:**

The paper proposes a pipeline for injecting knowledge from large language models into a robot instruction-following framework. Specifically, LLMs are used to select and sequence specific skills (i.e., actions) as to achieve the goal inferred from the language instruction.  The proposed architecture is evaluated in a real robotic system showing good performance at a variety of tasks, e.g., picking and place, object fetching, etc.

**Issues:**


In the first paragraph of Sec. 2, can you describe in detail what you mean by state, as in robot's observation or something more comprehensive that takes history into account or object relations, etc.?

Did you manually come up with each of the 551 skills or was there a procedure that does that?

Can you comment on to what extent these skills generalize to new objects and new configurations of objects not present in training set? It looks like the overall performance of the system is largely dependent on how well these skills work. Can you provide a subset of these in the paper? Also, are the 17 objects, 17 individual objects or are they object classes? Does that mean the robot would not do well if the objects were different from training time?

The related work is missing some works on grounding by Thomason and collaborators which is very closely related here:

Thomason, Jesse, et al. "Jointly improving parsing and perception for natural language commands through human-robot dialog." Journal of Artificial Intelligence Research 67 (2020): 327-374.
Thomason, Jesse, et al. "Language grounding with 3D objects." Conference on Robot Learning. PMLR, 2022.
Thomason, Jesse, et al. "Guiding exploratory behaviors for multi-modal grounding of linguistic descriptions." Proceedings of the AAAI Conference on Artificial Intelligence. Vol. 32. No. 1. 2018.

In particular, it would be interesting to comment in the paper on how the robot would be able to interpret words like "soft", or "heavy" that require non-visual perception.

It would also be good to discuss the pros and cons of the system in relation to more traditional approaches which parse language into logical form and use symbolic knowledge bases to infer what to do.

minor issues and comments.

With so many of the relevant details in appendices, this paper probably belongs in a journal.

In the last paragraph of the introduction, say exactly how many tasks, instead of "almost 100"

For the last sentence of paragraph 2 in Sec.2 to be true, I believe you would also need the discount factor to be 1.0 to make that equivalency.

**Quality Of The Limitations Section:**

Limitations are addressed clearly

**Reviewer Expertise:**

4: The reviewer is confident but not absolutely certain that the evaluation is correct

**Robotics Focus:**

Sufficient demonstration on hardware

**Strengths And Weaknesses:**


strengths:

The software integration effort is impressive.

The pipeline is creative and shows good performance on a variety of tasks

The evaluation on a real robot is a big plus.


weakness:

There is a lot of "devil is in the details" with the details either left out or in the appendix. This paper feel like it should be expanded to explain the details and go to a journal instead.




**Summary Of Recommendation:**


Overall, the paper is well written, the ideas are interesting and novel, and the work is demonstrated on a real robot. The only small drawback is that many details are missing and some comparisons with alternative approaches could be discuses better.

---

> ### Author Response · Authors · 2022-08-23
> **Thank you for your review, we have address the comments here and the paper**
>
> Thank you for your review and detailed feedback, which we believe has helped improve the paper.
>
> > There is a lot of "devil is in the details" with the details either left out or in the appendix. This paper feels like it should be expanded to explain the details and go to a journal instead.
>
> We agree that there are a lot of details in getting large robotic systems like this to work well. We have expanded our appendix to address additional details brought up and have appended the Appendix to the body of the main paper to allow for linking (we hope this makes it more searchable). If there are any additional details that you feel are not explained well we would be happy to address them. Regarding submission to CoRL versus a journal, we think that it would be unfortunate if the CoRL conference had to bias away from all papers that implement large robotic learning systems.
>
> We have also open sourced a simulated environment with a tabletop robot to help elucidate any details of SayCan ([Colab](https://colab.sandbox.google.com/github/saycan-corl/saycan-corl.github.io/blob/main/SayCan-Robot-Pick-Place.ipynb)).
>
> > In the first paragraph of Sec. 2, can you describe in detail what you mean by state, as in robot's observation or something more comprehensive that takes history into account or object relations, etc.?
>
> The state as referred to in Section 2 is quite general and can correspond to the robot’s state or the environment’s state, or any information required to compute affordances. We have added the following to Section 2 to clarify: “; where state can correspond to the robot, environment, or any information required to compute affordances”.  In this work, with our skills, the affordance functions use a state that consists of an image and robot base position.
> > Did you manually come up with each of the 551 skills or was there a procedure that does that?
>
> In the current form of the paper, we specified 17 objects and 7 different skill families to act on said objects (such as “pick”, “place”, “rearrange”) (discussed on L179 and Appendix Section B.3 and expanded discussion in Appendix Section C.2). As the system becomes more performant, we are planning to run it in a real environment to gather user-generated queries and use those to drive the skill acquisition process.
>
> > To what extent these skills generalize to new objects and new configurations of objects not present in training set? It looks like the overall performance of the system is largely dependent on how well these skills work. Can you provide a subset of these in the paper? Also, are the 17 objects, 17 individual objects or are they object classes? Does that mean the robot would not do well if the objects were different from training time?
>
> We agree with the reviewer that the overall performance of the system is dependent on the proficiency of the low-level skills. The skills presented in the paper are specific to 17 individual objects, such as a coke can, a bag of jalapeno chips, a water bottle, an apple, etc. These items are shown in Figure 3 and we have clarified in Appendix Section C.2, which now states “Each skill is thus an explicit text command performed by a low-level policy for that skill. For object above, we use the objects shown in Figure 3 (e.g., coke can, water bottle, jalapeno chips, apple, sponge, etc.). These objects can be placed in random configurations at locations throughout the scene. For location above, we consider several named locations with known positions shown in Figure 3 (e.g., table, trash can, etc.). This results in skills such as the following: …” and then a list of a few explicit skills.
>
> These skills generalize across object configurations and to some extent to different backgrounds and distractors. While SayCan is able to work with any low-level skill as long as it has an affordance function and language command associated with it, how to learn low-level skills efficiently and robustly is still an open problem.
>
> > Thomason et al.
>
> We appreciate the papers and agree that the body of work should be discussed, we have added them to the related work section.
>
> > Comment on how the robot would be able to interpret words like "soft", or "heavy" that require non-visual perception.
>
> We have found that the LLM in SayCan has semantic knowledge of non-visual properties. For example, if one asked for a “heavy item” it may bring the water, while asking for a “salty snack” may bring chips. The language model does the reasoning to semantically project any such command to useful skills. Appendix Listing 4 with chain of thought shows such an example where “spicy” prompts SayCan to bring “jalapeno chips”.
>
> The policies on the other hand have demonstration data with a variety of objects, and may learn those object’s unique properties. For example, we see varying grasps between sponges and coke cans. These variations may be a result of visual or non-visual properties and emerge from the data.

---

> > ### Author Response · Authors · 2022-08-23
> > **Continued**
> >
> > > It would also be good to discuss the pros and cons of the system in relation to more traditional approaches which parse language into logical form and use symbolic knowledge bases to infer what to do.
> >
> > We believe the primary benefit of SayCan compared to these works is in its flexibility and the generality of its parsing, at the cost of using a large model which may be expensive or slow to run. Semantic parsing often requires specific keywords that may be unnatural, require engineering effort to create the symbolic plans, or context specific training. A good example of an instruction that SayCan performs out of the box is “I’m hungry”, which then the robot brings a food item. Such a task is quite difficult for parsed language techniques. We have added the following statement and citations to the related work: “This enables a more flexible framework and more general, abstract tasks than semantic parsing [39, 42, 51].”
> >
> > > Minor comments
> >
> > Thank you for the comments. We have added ​”undiscounted” to L126 and have edited for additional comments.

---

### Meta-Review · Area_Chair_RMmR · 2022-08-15

**Recommendation:** Accept (Oral)
**Confidence:** 4

**Metareview:**

Below is a summary of the strengths and weaknesses of the paper, according to the reviewers.

Strengths:
- The overall proposed pipeline is interesting, simple, and novel, with very general applicability.
- The engineering and integration effort that has gone into this paper is very impressive.
- There is significant evaluation with a real robot.
- The video is very interesting and impressive.

Weaknesses:
- At a high level, all the tasks tested are quite similar (find object, pick up, go to location, put down).
- There are many small details which are missing from the paper and the appendix, which make it difficult to fully understand or re-implement this work.
- It is not clear what the definition of a "skill" is, as this is often defined differently in different papers.

In the rebuttal, please address the above weaknesses, as well as the other concerns and questions raised by the reviewers.

-------------

Update after the rebuttal:

All four reviewers have very high opinions of this paper. It is clearly a strong and timely paper, with a clear novel technical contribution which covers new ground at the intersection of NLP and robotics, and the paper has the potential to become a seminal paper in the field. The extensive real-world system put together is very impressive as a demo, and is an exciting step forwards towards robots assisting humans in everyday environments. The main weakness is the lack of reproducibility, due to (1) the large resources required to re-implement the method (although this is not strictly a limitation of the contribution), and (2) the lack of implementation details in the paper (although this can be addressed through the supplementary material, as reviewers have suggested). This is one of the major robot learning papers in recent months, and it will have broad appeal across the entire robot learning community.

**Best Paper Nomination:**

Yes

---

> ### Author Response · Authors · 2022-08-23
> **Drawer sequences, expanded appendix + open source, and skill definition**
>
> Thank you for your review and detailed feedback, which we believe has helped improve the paper.
>
> > At a high level, all the tasks tested are quite similar (find object, pick up, go to location, put down).
>
> Though many of the tasks in Section 4.1 are primarily built of this sequence, to show that our approach is not limited to these skills we include plans that require reasoning over embodiment (which may not require certain steps) and reasoning over drawer skills (Section 4.2, L273). Drawers require sequences such as "open drawer”, “pick object out of drawer”, “place object on counter”, “close drawer”, “pick object” and the reverse to place items into drawers. These drawer skills highlight both more complex plans and the extendability of the SayCan framework to accept new skills and sequences. We have expanded our discussion of such sequences in Section 4.2 and added a figure.
>
> > There are many small details which are missing from the paper and the appendix, which make it difficult to fully understand or re-implement this work.
>
> We appreciate the feedback and we have expanded the appendix and added details to address the questions raised. We have also reached out to the reviewers to be sure we add any additional missing details. To ensure re-implementability, we have also uploaded an open-source version of SayCan ([Colab](https://colab.sandbox.google.com/github/saycan-corl/saycan-corl.github.io/blob/main/SayCan-Robot-Pick-Place.ipynb)) with the tabletop environment shown in Appendix Section E and Figure 16.
>
> > It is not clear what the definition of a "skill" is, as this is often defined differently in different papers.
>
> Our definition of skill is an atomic text command that can be performed by the robot’s low-level policies. This definition is particularly natural for SayCan, because the designer can choose to specify any set of skills they want -- as long as each skill is described by a unique text command, the system can plan with it. In this work our skills are “pick up the <object>”, “put down the <object>”, “go to the <location>”, “find the <object>”, and several drawer-related commands, while for the open source tabletop setting our skills are “pick up the <object> and place it in/on the <location/object>”. We have expanded upon our discussion in Section 3 and Appendix Section C.2 to complement existing discussions (L41, L66, L138):
> * L179: “Inspired by common skills one might pose to a robot in a kitchen environment, we propose 551 skills that span seven skill families and 17 objects. The skill families are picking, placing, and rearranging objects, placing objects in specific configurations, opening and closing drawers, and navigating to various locations. A skill may then be "pick up the apple" or "go to the trash can".”
> * L702: “Each skill is thus an explicit text command performed by a low-level policy for that skill. For object above, we use the objects shown in Figure 3 (e.g., coke can, water bottle, jalapeno chips, apple, sponge, etc.). These objects can be placed in random configurations at locations throughout the scene. For location above, we consider several named locations with known positions shown in Figure 3 (e.g., table, trash can, etc.). This results in skills such as the following: …” and then a list of a few explicit skills.